# GPT Shortcuts: Learning Iterative Text Generation Patterns from a Dialogue

## Abstract

LLM-powered conversational interfaces (e.g., ChatGPT, Claude, and Gemini) support iterative text generation, enabling users to easily generate tailored texts (e.g., texts that should address domain-specific constraints) through a series of follow-up text editing requests. However, generating such tailored texts that address the user-specified constraints across multiple different contexts requires repetitive text generation efforts, which is cumbersome, inefficient, and demanding. To address this challenge, we introduce the concept of *GPT shortcuts*, which is designed to 1) learn iterative text generation patterns from a dialogue and 2) apply these learned patterns to *directly* generate the tailored text. GPT shortcuts generate texts that address necessary constraints while maintaining similar structural appearance to the target text in the dialogue, across different contexts. To assess the capability of language models in generating GPT shortcuts, we present SHORTCUTBENCH, a benchmark consisting of 250 crowdsourced iterative text generation dialogues across five text generation tasks. Using SHORTCUTBENCH, we conducted an analysis using six LLMs and four prompting methods, varying ways to specify necessary constraints to address in the prompt. We found that 1) larger models generally outperform smaller models, 2) self-explanatory constraints within the target text are effective, and 3) precisely specifying necessary constraints to address is critical for improving the performance.[1]

## 1 Introduction

Large Language Models (LLMs) have been utilized for various types of text generation tasks (Lin et al., 2024). Users can generate texts addressing varying constraints, such as domain-specific requirements (e.g., writing a clinical report following specific diagnostic rules (Wang et al., 2024)) and structure of the text (e.g., including "unusual observations" and "treatment plan" in the clinical report as a numbered list), depending on their context and preferences. To generate such tailored texts with conversational LLMs (e.g., ChatGPT, Claude, and Gemini), users refine and revise the generated texts through multiple turns until they get the desired texts (Figure 1-a). Unlike single-turn generation, conversational text generation offers an easy and interactive way to create texts through incremental revisions.

However, iterative text generation poses significant challenges to users when repetitively generating tailored text across different contexts (e.g., writing clinical reports that address hospital-specific medical guidelines in a consistent format for multiple patients). It requires a case-by-case review and revision process to address them, which is cumbersome, time-consuming, and demanding. These challenges can be mitigated by allowing users to *directly* generate target texts that address the constraints specified in the iterative text generation dialogue, which we represent as a *shortcut*. The shortcut allows users to bypass the need for repeatedly specifying the desired constraints, streamlining future text generation tasks. Despite the potential, the concept of shortcut in the context of the iterative text generation has not been explored yet.

We introduce a GPT shortcut generation task, aiming at generating a function that directly generates texts resembling the target text in an iterative text generation dialogue (Figure 1-b). Specifically, the generated text should address the user-specified constraints while maintaining a similar appearance

---

[1]Demo: http://43.202.150.120:5173/

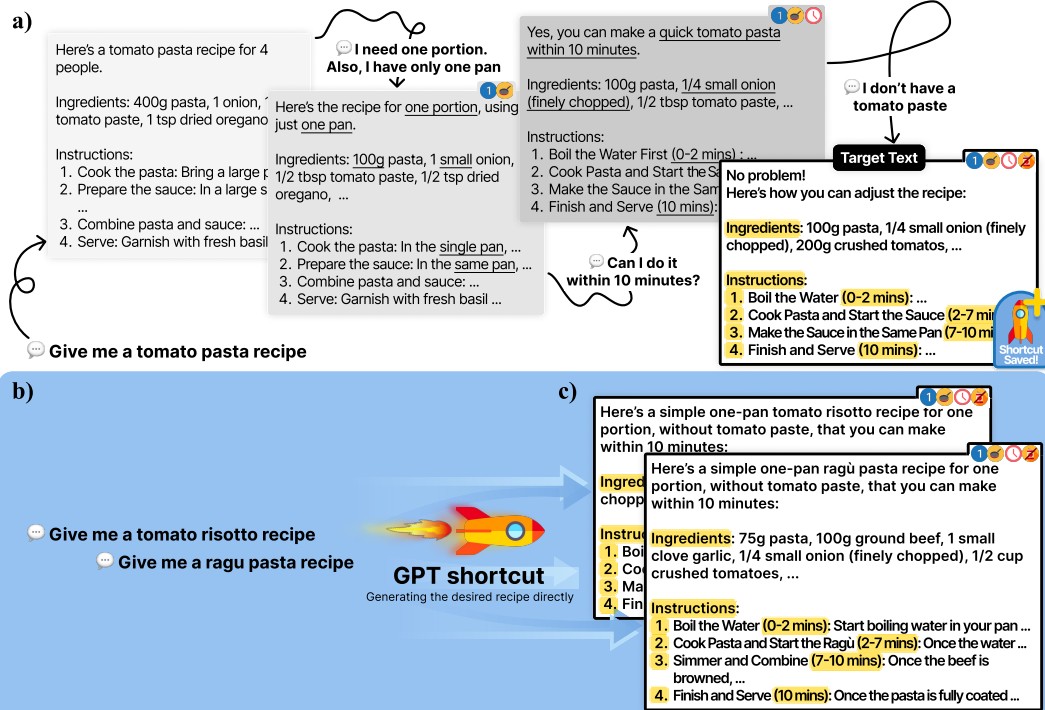

Figure 1: A visual illustration of the GPT shortcut. a) Initially, the target text ( Target Text ) is generated through iterative revisions of LLM-generated texts through a multi-turn dialogue. In this case, the user requests a tomato pasta recipe tailored to their circumstances where the constraints were not immediately obvious at the first place but emerged after reading the generated text. The GPT shortcut can be created from the process, learning the iterative text generation patterns. b) The GPT shortcut *directly* produces the desired texts for different inputs (e.g., "*Give me a tomato risotto recipe*" or "*Give me a ragu pasta recipe*"). c) In this example, the GPT shortcut directly generates personalized recipes for tomato risotto and ragu pasta. The generated recipes address the user-specified constraints ( ) and have similar structural appearance to the target text. GPT shortcuts streamline the text generation process, offering easier, more efficient, and more reliable experience compared to manual iterations.

and structure to the target text (Figure 1-c). The GPT shortcut generation task requires challenging steps, namely 1) identifying user-specified constraints to address from the iterative text generation dialogue and 2) developing text generation pipelines that *reliably* generate the texts resembling the target text. This paper explores the capability of off-the-shelf LLMs in performing the GPT shortcut generation task to understand the limitations and gain insights into developing novel methods.

Since there are no existing benchmarks for evaluating GPT shortcut generation capabilities, we introduce SHORTCUTBENCH, a benchmark dataset that consists of 250 crowdsourced iterative text generation dialogues across five text generation tasks. SHORTCUTBENCH includes 1) iterative text generation dialogues using an input text (e.g., a dialogue that produces a news article summary), 2) three different input texts per task as test cases (e.g., alternative news articles to summarize), and 3) checklists representing necessary constraints to address in the generated text (e.g., the summary of a news article should contain the timeline). The performance of GPT shortcuts has been computed by two metrics: 1) $\text{SB}_{\text{con}}$, measuring an average proportion of the necessary constraints addressed, and 2) $\text{SB}_{\text{app}}$, measuring the similarity in structural appearance between the generated text and target text. $\text{SB}_{\text{con}}$ was computed by prompting GPT-4o[2] as a judge, assessing whether each of the necessary constraints has been addressed. $\text{SB}_{\text{app}}$ was computed by comparing the sequence of line lengths between the generated text and the target text.

---

[2] `gpt-4o-2024-05-13` was used.

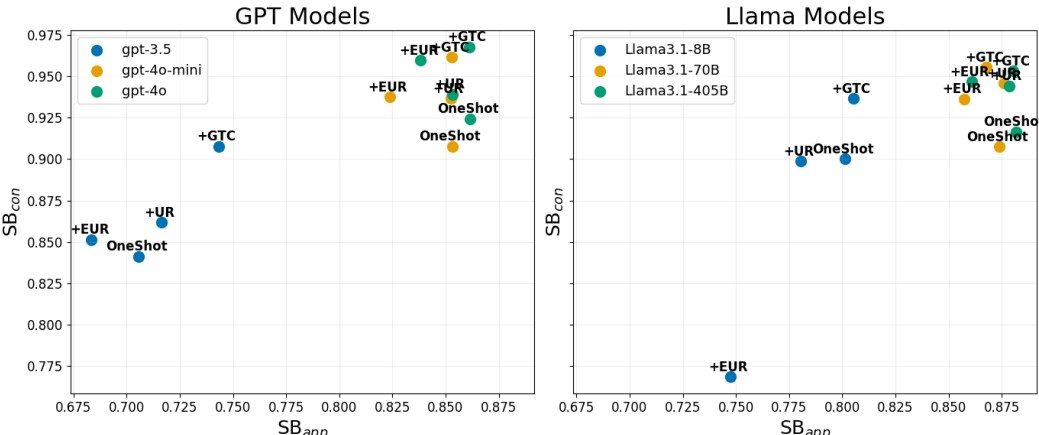

Figure 2: The performance of six LLMs and four prompting methods. Overall, larger models outperformed smaller models. **+GTC** consistently achieved the best $SB_{con}$ score and comparable $SB_{app}$ score with other methods. It highlights the importance of precisely capturing the constraints, a representation of the iterative text generation pattern. **+EUR** did not improve both $SB_{con}$ and $SB_{app}$ scores compared to **+UR**, highlighting the limitation of Chain-of-Thought approach in capturing and applying the patterns.

Using SHORTCUTBENCH, we conducted an ablation study using six off-the-shelf LLMs, GPT-{3.5-turbo, 4o-mini, 4o} and Llama3.1-{8B, 70B, 405B}. We compared four one-shot prompting methods: prompting with the target text as an example output only (**OneShot**), with the list of user utterances in the dialogue as explicit constraints in addition to the target text (**+UR**), with Chain-of-Thought instructions for identifying necessary constraints to address in the user utterances (**+EUR**), and with the ground-truth constraints (**+GTC**). Figure 2 shows an overall performance of the LLMs and methods. Overall, larger models outperformed smaller models for both $SB_{con}$ and $SB_{app}$, illustrating the requirements of reasoning capability (Table 2). Small models, in particular, struggled to generate texts with similar appearance (Figure 5). One-shot prompting was effective when the example output (i.e., the target text) is self-explanatory, clearly revealing the necessary constraints. However, it struggled with complex dialogues and example outputs that lack such explanations. Providing explicit constraints in addition to the example significantly improved $SB_{con}$ in these cases (Table 3). Moreover, specifying precise constraints (i.e., correct constraints to address) in the prompt further improved $SB_{con}$ with comparable $SB_{app}$. However, the Chain-of-Thought approach (Wei et al., 2022) for precisely identifying the necessary constraints from the user utterances did not improve $SB_{con}$ and even degraded $SB_{app}$ (Table 2). To sum up, our findings suggest that LLMs can apply the iterative text generation patterns to different contexts, and precisely capturing these patterns is important for improving performance, especially for small models.

GPT shortcuts can provide practical value by streamlining the iterative text generation process, allowing users to easily create task automation that aligns with their specific needs by simply demonstrating the text generation process. This can be seen as a Programming-by-Demonstration approach (Cypher & Halbert, 1993; Cambronero et al., 2023) within prompt programming (Liang et al., 2024; Beurer-Kellner et al., 2023), which has been recognized as an effective approach for end-users to create customized task automation (Li et al., 2017a;b). For example, by demonstrating how to write clinical summaries in specific cases, medical professionals can create GPT shortcuts that produce consistent summaries for different patients, addressing complex regulatory requirements and including necessary medical terminologies (Van Veen et al., 2023; Lee et al., 2023). Developers can leverage GPT shortcuts to produce consistent code reviews that follow the company-specific coding standards and review formats (Yang et al., 2024; Lu et al., 2023), by demonstrating the code review generation for specific code. We believe that our work represents a promising starting point for enabling the complex yet powerful task automation by formalizing the problem as learning iterative text generation patterns and evaluating the capabilities of LLMs.

To sum up, this paper makes the following contributions:

- Introducing an important yet underexplored task of learning iterative text generation patterns from a user dialogue — GPT shortcut generation task.

- SHORTCUTBENCH, a benchmark dataset that consists of 250 iterative text generation dialogues across five text generation tasks, designed to evaluate the capability of language models in generating GPT shortcuts, along with two metrics $SB_{con}$ and $SB_{app}$.

- Empirical results showing the performance of the GPT shortcut generation task across six language models and four prompting methods, highlighting the effectiveness of self-explanatory constraints within the target text, the benefits of specifying explicit constraints especially for complex dialogues, and the importance of precisely identifying constraints in the dialogue.

## 2 RELATED WORK

**Conversational LLMs.** LLM-powered conversational agents have made significant success in addressing diverse text generation needs (Achiam et al., 2023). Despite the ease of use and flexibility of the conversational interaction (Flohr et al., 2021), research has shown that novice users face challenges in creating effective prompts to generate the desired texts (Zamfirescu-Pereira et al., 2023; Kim et al., 2023; Dang et al., 2022). Iterative text generation allows users to more easily create the desired texts by a step-by-step approach, prompting LLMs to further address constraints (Wen et al., 2024) in multi-turn dialogues. Despite the needs of consistently addressing the constraints specified in the dialogue, applying the iterative text generation patterns in a different context remains unexplored. We address the challenge by introducing the concept of GPT shortcut, which learns the iterative text generation patterns and applies them to a different context.

**Evaluating LLM capabilities.** Research has extensively evaluated the capabilities of LLMs in various task categories, including knowledge retrieval (Vu et al., 2023; Chen et al., 2024), question and answering (Zhuang et al., 2023; Alonso et al., 2024), reasoning (Rein et al., 2023; Liu et al., 2024), evaluation (Zheng et al., 2023; Zhou et al., 2024), and novel idea generation (Si et al., 2024; Radensky et al., 2024). However, most benchmarks focus on single-turn interactions where a prompt is given and LLM response is assessed. Recently, WILDBENCH (Lin et al., 2024) examined performance of LLMs in supporting challenging tasks using real multi-turn dialogues (Zhao et al., 2024), but it is still limited as a benchmark for the GPT shortcut generation task in terms of dialogue complexity (89% has $\leq 2$ turns) and coherency across tasks. To evaluate LLM capabilities in GPT shortcut generation task through more complex dialogues, we introduce SHORTCUTBENCH, a crowdsourced benchmark that consists of 250 iterative text generation dialogues ($\geq 3$ turns) across five text generation tasks.

**Identifying input-output relations.** Research has investigated methods for effectively identifying the relationship between an input and output text. Existing approaches include prompting LLMs to describe these relationships (Honovich et al., 2022) and optimizing prompts by sampling (Shin et al., 2020; Zhou et al., 2022). The GPT shortcut generation task can be seen as identifying the relationship between the input and output text by analyzing the iterative text generation dialogue, which involves more practically useful and complex patterns compared to simpler relationships discussed in existing benchmarks, such as negating the input sentence (Honovich et al., 2022). As the first step, we evaluate the capability of LLMs in capturing and applying the iterative text generation patterns using SHORTCUTBENCH.

## 3 GPT SHORTCUT TASK

We define the GPT shortcut generation task as follows: Given a user dialogue $d$, which demonstrates iterative text generation producing a target text $y$ when given input $x$ ($d : x \rightarrow y$), a GPT shortcut is defined as $f_d(x')$, which directly generates target text $y'$ for a new input $x'$, where $d : x' \rightarrow y'$. Figure 1-a) shows a dialogue $d$ that generates target text $y$ from the input $x$, and Figure 1-b) demonstrates the GPT shortcut $f_d(x')$ directly generating target text $y'$ from given a new input $x'$. GPT shortcuts enable users to perform the text generation tasks in an easy, efficient, reliable way while avoiding repetitive conversations in varying contexts.

GPT shortcuts can be implemented in different ways such as code (Cai et al., 2023), a prompt (Honovich et al., 2022; Zhou et al., 2022), and a chain of prompts (Wei et al., 2022; Wu et al., 2022), which ensures generating texts that meet necessary constraints in the user dialogue while maintaining

Table 1: Statistical comparison between SHORTCUTBENCH and WILDBENCH$_{\text{Editing}}$, a subset of WILDBENCH filtered for dialogues tagged "Editing" in either the primary or secondary categories. SHORTCUTBENCH has more dialogues and turns, which is better suited for GPT shortcut generation tasks requiring multi-turn dialogues. Also, SHORTCUTBENCH focuses on five text generation tasks with clear goals while WILDBENCH$_{\text{Editing}}$ consists of less coherent tasks with different goals.

| Dataset | # Dialogues | # Turns | | | # Tasks | # Metrics |
|---------|-------------|---------|-----|--------|---------|-----------|
| | | Sum | Avg | Median | | |
| WILDBENCH$_{\text{Editing}}$ | 124 | 340 | 2.74 | 2 | 8 | 1 (Checklists) |
| SHORTCUTBENCH | 250 | 2094 | 8.37 | 8 | 5 | 2 (Checklists, Appearance) |

the structure of the target text. In this paper, we focus on the prompting methods to implement GPT shortcuts and evaluate the performance of the methods.

## 4 SHORTCUTBENCH

SHORTCUTBENCH is a benchmark dataset for assessing the capability of language models in the GPT shortcut generation task, which consists of 1) 250 iterative text generation dialogues across 5 text generation tasks, 2) a source text (for creating GPT shortcuts) and three candidate texts (for testing GPT shortcuts) for each dialogue, and 3) checklists for assessing whether the generated text from each of the candidate texts addresses necessary user-specified constraints in the dialogue.

### 4.1 DATA COLLECTION PROCEDURE

Our goal was to collect iterative text generation dialogues via crowdsourcing so that the dataset captures realistic and diverse human intents and behavior in iterative text generation. We selected five tasks that 1) crowd workers can easily engage in without any significant barrier and 2) have been widely studied in NLP: text summarization (Nallapati et al., 2016), text simplification (Surya et al., 2018), essay grading (Wang et al., 2022b), story generation (Fan et al., 2018), and QA (Fan et al., 2019). For each task, we chose one dataset to select a source text (for creating GPT shortcuts) and three candidate texts (for testing GPT shortcuts). Then we collected 50 iterative text generation dialogues for each task via crowdsourcing on the Prolific platform. To collect multi-turn dialogues, we asked each participant to improve the generated text by their preferences at least two times so that the collected dialogues contain at least three user utterances. Specifically, participants were required to select at least three text generation outcomes that they liked and considered to have improved over the turns. More detailed task selection process, algorithms for choosing the source and candidate texts, and the data collection interface are included in Appendix A.

With the collected iterative text generation dialogues, we generated checklists for each dialogue, which captures necessary constraints (e.g., "Include timeline on the summary.") that the target text addresses. It is important to note that not all the user-specified constraints in the dialogue have been addressed in the target text. To make sure that the checklists cover the addressed constraints, we took a two-step approach. First, we identified all the requested constraints. Specifically, we leveraged GPT-4o to generate all the user-specified constraints from the dialogue, similar to prior approaches to creating such checklists (Lin et al., 2024). Then we manually reviewed the checklists by examining their coverage and generalizability to make sure that the checklists are not too specific to the source text but applicable to other input texts (e.g., for a story generation task where the source story contains DVDs, we revised a constraint *"make DVDs antique"* into *"make a commonly used item in the past become an antique"*). Second, we evaluated whether the target text addresses each of the constraints using GPT-4o as a judge. We conducted the evaluation ten times for each constraint and filtered constraints that have been determined as addressed at least seven times.

### 4.2 STATISTICS

Table 1 shows the statistics of SHORTCUTBENCH and WILDBENCH$_{\text{Editing}}$, a subset of WILDBENCH (Lin et al., 2024), filtered to include dialogues tagged with "Editing" in either the primary or secondary categories. To the best of our knowledge, WILDBENCH$_{\text{Editing}}$ is the only benchmark that

includes multi-turn text editing dialogues. However, it is not sufficient for evaluating the capability of GPT shortcut generation tasks due to the limited number of turns (2.7 on average, counting both the user and assistant utterances). Furthermore, the goal of the dialogue is not necessarily focused on iteratively generating a single text, which doesn't align with the goal of evaluation. SHORTCUTBENCH, on the other hand, consists of 250 dialogues focused on iterative text generation, enabling clear evaluation of the GPT shortcut generation capability. Figure 3 compares the turn distributions. SHORTCUTBENCH

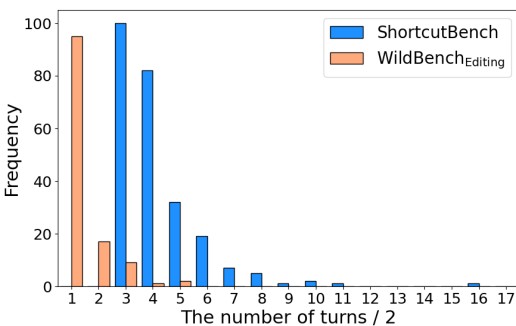

Figure 3: Comparison of the turn distribution.

consists of dialogues with more number of turns, demonstrating more complex iterative text generation patterns. By focusing on five distinct text generation tasks, the evaluation can also highlight how GPT shortcut generation techniques account for the specific nature of each task.

### 4.3 METRICS

Since the goal of GPT shortcuts is to generate texts resembling the target text in the dialogue, the generated texts are expected to not only address the necessary constraints in the dialogue but also to be structurally similar to the target text. SHORTCUTBENCH computes the overall performance by taking an average of two scores: $\text{SB}_{\text{con}}$ and $\text{SB}_{\text{app}}$.

$\text{SB}_{\text{con}}$ is the proportion of necessary constraints addressed, which is computed by

$$\text{SB}_{con} = \frac{\text{The number of necessary constraints addressed}}{\text{The total number of necessary constraints to address}}$$

where the necessary constraints refer to a set of user-specified constraints that have been addressed in the target texts. SHORTCUTBENCH includes the checklists to verify each of the necessary constraints.

$\text{SB}_{\text{app}}$ measures to what extent the generated text structurally resembles the target text. One of the reasons for including $\text{SB}_{\text{app}}$ is to account for *implicit* constraints that the user has not explicitly requested but are present in the target text. For instance, when asked to grade a student essay, the output may include bullet points outlining grading criteria, even though the user did not explicitly ask for this. Since the user may prefer such structured texts over a plain text in future generations, we use $\text{SB}_{\text{app}}$ to incorporate these implicit preferences into the evaluation, even when the input differs. However, we recognize that generating texts with similar formats may not always be desirable, depending on the use cases of GPT shortcuts.

We compare the sequence of line lengths between the generated text and target text. Specifically, we compute the normalized value of the minimum edit distance between the sequences where the edit distance is computed by

$$\text{SED}(a, b) = \sum_{i=1}^{|a|} |a_i - b_i| + \sum_{i=|a|+1}^{|b|} b_i$$

where $a$ and $b$ are the sequence of line lengths and $|a| \leq |b|$. The minimum edit distance can be computed by Dynamic Programming where the recurrence relation is defined by

$$\text{MSED}(i, j) = \begin{cases} \text{MSED}(i-1, j-1) & \text{if } a_i = b_j \\ \min \begin{cases} \text{MSED}(i, j-1) + b_j, \\ \text{MSED}(i-1, j) + a_i, \\ \text{MSED}(i-1, j-1) + |a_i - b_j| \end{cases} & \text{otherwise} \end{cases}$$

where MSED($i, j$) denotes the minimum sequence edit distance between two sequences $(a_1, a_2, ..., a_i)$ and $(b_1, b_2, ..., b_j)$. Finally, SB$_{con}$ is computed by normalizing the minimum sequence edit distance as follows.

$$\text{SB}_{app} = 1 - \frac{\text{MSED}(|g|, |t|)}{\sum g + \sum t}$$

where $g$ and $t$ are the sequence of line lengths of the generated text and target text, respectively.

## 5 METHODS FOR GENERATING GPT SHORTCUTS

Conceptually, GPT Shortcuts are functions that produce a target text from a set of input texts. Identifying potential input texts within the iterative text generation dialogue requires analyzing the dialogue along with the task demonstration and the target text. In this paper, we consider a simple case where the function takes a single input text. The following methods were used to generate GPT shortcuts. See Appendix B for the prompts used for the methods.

### 5.1 TEXT GENERATION METHODS

**OneShot.** Given the input text and target text in the iterative text generation dialogue, one-shot learning can produce the desired text for another input text. This is a simple but effective approach, given its remarkable performance and ability to generate texts similarly formatted to the target text. To produce the desired text, LLMs are expected to address user-specified constraints that have been implicitly presented in the target text.

**OneShot+UserRequests (+UR).** A limitation of **OneShot** is the lack of information about task demonstrations. Therefore, we provide the list of user utterances in the dialogue to illustrate how the target text has been generated from the input text. LLMs are expected to address *effective* user requests, which is a subset of the list of user requests that have been addressed in the target text.

**OneShot+EffectiveUserRequests (+EUR).** Naively putting the entire list of user requests could confuse LLMs as only the subset of user requests should be addressed. Using the Chain-of-Thought approach (Wei et al., 2022), we prompted LLMs to (1) evaluate whether each user request is addressed in the target text, (2) list the effective user requests, and (3) produce an output text that addresses the identified effective user requests. In this way, we expect LLMs can be less confused about which user requests to address.

**OneShot+GroundTruthConstraints (+GTC).** We put the ground-truth constraints in the checklists in the prompt, expecting to yield the ceiling performance of the other methods as it clearly informs correct constraints without any confusion.

### 5.2 LANGUAGE MODELS

We evaluated six off-the-shelf language models for the GPT shortcut generation task: GPT-{3.5-turbo, 4o-mini, 4o} and Llama3.1-{7B, 70B, 405B} to assess the general performance and the impact of scales.

## 6 EVALUATION RESULTS

In general, larger models outperformed smaller models for both SB$_{con}$ and SB$_{app}$ scores. Table 2 shows the overall performance with SB$_{con}$ and SB$_{app}$ scores for the language models and methods where the overall score was computed by taking the average of the two scores. **+UR** outperformed **OneShot**, but **+EUR** showed worse performance than **+UR**. Inspecting the performance, **+EUR** achieved a lower SB$_{app}$ score with a comparable SB$_{con}$ score, compared to **+UR**. We observed that the Chain-of-Thought approach of **+EUR** often produced lengthy texts compared to the target texts. Llama3.1-8B showed a significant performance reduction on **+EUR**, which aligns with the findings that CoT prompting is not effective for small models (Wei et al., 2022). For the overall score, **+GTC** *consistently* achieved the best overall scores across all the language models. **OneShot** achieved the

Table 2: The overall performance of the six LLMs and four methods. The overall score was computed by taking the average of $SB_{con}$ and $SB_{app}$ scores. The $SB_{con}$ and $SB_{app}$ scores represent the average scores of the five text generation tasks (see Appendix D for the scores for each text generation task). The bold phased and underlined scores are the best score and worst score between the methods, respectively.

| Metric | Method | GPT Models | | | Llama Models | | |
| --- | --- | --- | --- | --- | --- | --- | --- |
| | | GPT-3.5 | GPT-4o-mini | GPT-4o | Llama3.1-8B | Llama3.1-70B | Llama3.1-405B |
| Overall | OneShot | 0.773 | 0.880 | 0.893 | 0.851 | 0.891 | 0.899 |
| | +UR | 0.789 | 0.894 | 0.896 | 0.840 | 0.911 | 0.911 |
| | +EUR | 0.767 | 0.881 | 0.899 | 0.758 | 0.897 | 0.904 |
| | +GTC | **0.826** | **0.907** | **0.914** | **0.871** | **0.912** | **0.917** |
| $SB_{con}$ | OneShot | 0.841 | 0.908 | 0.924 | 0.900 | 0.907 | 0.916 |
| | +UR | 0.862 | 0.936 | 0.939 | 0.899 | 0.946 | 0.944 |
| | +EUR | 0.851 | 0.937 | 0.960 | 0.768 | 0.936 | 0.947 |
| | +GTC | **0.908** | **0.961** | **0.968** | **0.936** | **0.955** | **0.953** |
| $SB_{app}$ | OneShot | 0.706 | **0.853** | **0.862** | 0.801 | 0.874 | **0.882** |
| | +UR | 0.716 | 0.852 | 0.853 | 0.780 | **0.876** | 0.878 |
| | +EUR | 0.683 | 0.824 | 0.838 | 0.747 | 0.857 | 0.861 |
| | +GTC | **0.743** | **0.853** | 0.861 | **0.805** | 0.868 | 0.880 |

worst overall scores, particularly low $SB_{con}$ scores, but performed well in terms of $SB_{app}$ scores. The detailed $SB_{con}$ and $SB_{app}$ scores for each of the text generation tasks are included in Appendix D. We report the following in-depth observations.

**Including explicit constraints in the prompt significantly helps, especially for complex dialogues.** **OneShot** achieved reasonably high $SB_{con}$ score ($\geq 0.9$ except for GPT-3.5), which implies that LLMs are able to capture the necessary constraints just with the target output as an example. As a possible reason, we observed that the example output is often self-explanatory. For example, LLM responses tend to contain an introductory statement that explains the text to generate, such as *"Sure, here are the key points you should remember for your test!"*, which clearly describes the necessary constraints. Also, structured texts like a numbered list with headers also specifically guide which kinds of texts to generate at each point. However, **OneShot** significantly underperformed compared to other methods when the target text did not clearly imply the necessary constraints. For instance, the gap of $SB_{con}$ scores between **OneShot** and **+GTC** was 4.5 times larger on average when the dialogue contained more than four constraints compared to fewer (Table 3). The performance gap was the largest in the story generation task, likely because the generated stories tend to be long texts consisting of multiple paragraphs without such explicit cues. This highlights the importance of comprehensively *presenting* the necessary constraints, especially for complex text generation tasks.

**Precise constraints in the prompt improve $SB_{con}$ scores with comparable $SB_{app}$ scores.** For all the language models, **+GTC** achieved higher $SB_{con}$ score than both **+UR** and **+EUR**. It suggests that precisely identifying the necessary constraints in the dialogue is important. **+EUR** improved the overall performance a bit for large models (GPT-4o and Llama3.1-405B), but for the other models the overall performance degraded. Notably, GPT-3.5 achieved a significant gain on $SB_{con}$ with **+GTC** compared to **+UR** when the target text does not imply the necessary constraints (i.e., a dialogue with more than four constraints and the story writing task). It highlights the potential of running GPT shortcuts using the small models as 'tool users' while using more powerful models for generating GPT shortcuts as 'tool makers' (Cai et al., 2023). For the appearance, **+GTC** consistently achieved comparable $SB_{app}$ scores with **OneShot** for all the language models while **+UR** and **+EUR** generally hurt the score. Even **+GTC** improved $SB_{app}$ score for GPT-3.5 compared to **OneShot**. It suggests that less precise constraints negatively impact the appearance of the generated text.

**Small models struggle to generate texts with similar appearance.** Figure 5 shows the cumulative distribution of $SB_{app}$ for each model. In general, the cumulative lines become steeper for the larger model, which means that larger models are better at generating texts with similar appearance while addressing the necessary constraints. Small models (GPT-3.5 and Llama3.1-8B) showed the cumulative percentage 25.2% and 14.33% on average at $SB_{app} = 0.6$ where the appearance of target text and generated text are significantly different (see Appendix C for the examples). We also observed notable differences in the appearance of generated texts between the small models. GPT-3.5

Table 3: $SB_{con}$ scores for simple dialogues (# constraints $\leq 4$), complex dialogues (# constraints $\geq 5$), four text generation tasks (excluding the story writing task), and story generation task. The performance of the four text generation tasks was aggregated by taking the average. We used boldface for delta scores that are five times larger than the delta of the other case (i.e., Simple vs. Complex, Non-Story vs. Story). The detailed scores are included in Appendix D.

| Method | GPT Models | | | Llama Models | | |
|---|---|---|---|---|---|---|
| | GPT-3.5 | GPT-4o-mini | GPT-4o | Llama3.1-8B | Llama3.1-70B | Llama3.1-405B |
| **Simple Dialogue** | | | | | | |
| OneShot | 0.874 | 0.938 | 0.950 | 0.905 | 0.930 | 0.940 |
| +UR | 0.909 | 0.933 | 0.947 | 0.917 | 0.955 | 0.958 |
| +GTC | 0.907 | 0.957 | 0.964 | 0.928 | 0.952 | 0.949 |
| Delta (GTC-OneShot) | +0.033 | +0.019 | +0.014 | +0.023 | +0.022 | +0.009 |
| Delta (GTC-UR) | -0.002 | +0.024 | +0.017 | +0.011 | -0.003 | -0.009 |
| **Complex Dialogue** | | | | | | |
| OneShot | 0.774 | 0.861 | 0.907 | 0.912 | 0.899 | 0.893 |
| +UR | 0.781 | 0.899 | 0.937 | 0.900 | 0.951 | 0.933 |
| +GTC | 0.910 | 0.965 | 0.967 | 0.956 | 0.955 | 0.973 |
| Delta (GTC-OneShot) | +0.136 | **+0.104** | +0.060 | +0.044 | +0.056 | **+0.080** |
| Delta (GTC-UR) | **+0.129** | +0.066 | +0.030 | **+0.056** | +0.004 | **+0.040** |
| **Non-Story Generation** | | | | | | |
| OneShot | 0.879 | 0.930 | 0.941 | 0.899 | 0.921 | 0.933 |
| +UR | 0.896 | 0.943 | 0.938 | 0.887 | 0.947 | 0.944 |
| +GTC | 0.908 | 0.961 | 0.965 | 0.934 | 0.954 | 0.946 |
| Delta (GTC-OneShot) | +0.029 | +0.031 | +0.024 | +0.035 | +0.033 | +0.013 |
| Delta (GTC-UR) | +0.012 | +0.018 | +0.027 | +0.047 | +0.007 | +0.002 |
| **Story Generation** | | | | | | |
| OneShot | 0.691 | 0.817 | 0.857 | 0.906 | 0.852 | 0.852 |
| +UR | 0.724 | 0.910 | 0.944 | 0.945 | 0.940 | 0.944 |
| +GTC | 0.907 | 0.965 | 0.980 | 0.947 | 0.961 | 0.980 |
| Delta (GTC-OneShot) | **+0.216** | **+0.148** | **+0.123** | +0.041 | +0.109 | **+0.128** |
| Delta (GTC-UR) | **+0.183** | +0.055 | +0.036 | +0.002 | +0.021 | **+0.036** |

often generated significantly shorter texts (e.g., generating only three sentences when the target text in the example spans more than ten paragraphs) whereas Llama3.1-8B often generated overly long texts. Llama3.1-8B occasionally showed odd text generation behavior, such as producing texts that were not properly finished, repeating a few sentences over and over.

**Llama3.1-8B outperformed the other language models for the story generation task.** Figure 4 shows the overall performance on the story gen-
eration task. In **OneShot**, Llama3.1-8B signifi-
cantly outperformed the other language models,
showing the largest performance gap compared
to the other methods. It implies that Llama3.1-
8B could capture important features in a story
example better than larger models. It is worth
further investigating why Llama3.1-8B can be
comparable, when properly optimized (Lin et al.,
2024), and even outperform larger models in
creative tasks. The gap of $SB_{app}$ score was
larger than that of $SB_{con}$ between Llama3.1-8B
and Llama3.1-{70B, 405B}. GPT-3.5**+GTC**
performed significantly worse than Llama3.1-
8B**+OneShot**, likely due to GPT-3.5's tendency
to generate overly short texts (e.g., generating
only a two-sentence story while the example
story consists of more than three paragraphs).

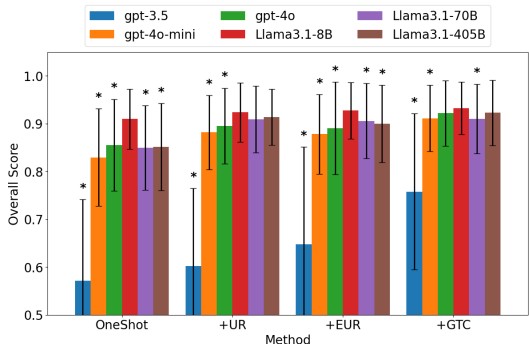

Figure 4: The overall performance for the story generation task, along with statistical test results (Welch's t-test (Welch, 1947), p<0.01) comparing Llama3.1-8B with other models using the same method. The detailed scores including $SB_{con}$ and $SB_{app}$ are included in Appendix D.

## 7 CONCLUSION AND FUTURE WORK

**Conclusion.** We introduced the concept of GPT shortcuts, which learns iterative text generation patterns from a dialogue so that users can directly generate the target text in the dialogue in different contexts. Our evaluation highlights the capabilities and limitations of the language models in generating GPT shortcuts. As the iterative text generation through conversational LLMs is a natural user interaction, there exists diverse user needs and iterative generation patterns. Developing robust methods for learning such diverse and complex patterns could serve as a foundational step

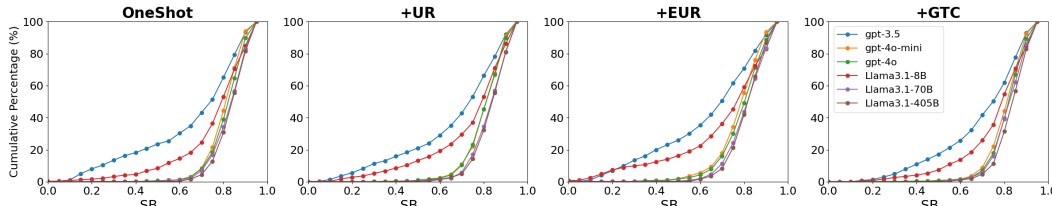

Figure 5: Cumulative distribution of $SB_{app}$ scores for each method and model. Models with steeper lines generate texts with closer structural appearance of target text. The distribution shows that small models struggle to generate texts with the closer appearance.

towards more efficient generation of complex texts, enabling new applications in domains with specialized requirements, such as technical writing, education, marketing, and law.

**Limitations and future work.** This work focused on five text generation tasks that have been actively investigated in the NLP community, but considering more tasks in evaluation could inform broader perspectives about the performance of GPT shortcuts. Also, we believe that considering more complex iterative text generation dialogues can offer richer insights into how language models capture the complex patterns, but collecting complex dialogues at scale is challenging. Designing a crowdsourcing workflow for collecting complex iterative text generation dialogues could be interesting future work. Starting from the crowdsourced utterances in SHORTCUTBENCH, synthetically generating complex dialogues is another promising approach. Finally, future work could develop novel GPT shortcut generation methods by effectively capturing and applying the iterative text generation process.

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

APPENDIX

# A    DATA COLLECTION

## A.1    TASK SELECTION PROCESS FOR THE EVALUATION

Table 4: A list of tasks identified in the task selection process. Tasks with bold phase were selected for evaluation.

| Meta-category | Task category (Dataset) |
|---|---|
| QA | **Question Answering** (*Data generated using GPT-4*)
Mathematics |
| Summarization | **Summarization** (CNN/Daily Mail Dataset (Nallapati et al., 2016))
Title Generation |
| Translation | **Text Simplification** (PubMed 20K (Dernoncourt & Lee, 2017))
Data to Text |
| Evaluation | **Text Quality Evaluation** (ASAP Dataset (Hamner et al., 2012))
Answer Verification |
| Creation | **Story Completion** (ROCStories (Mostafazadeh et al., 2016))
Dialogue Generation |

We carefully selected five text generation tasks from the following process. We started with a list of NLP task types in prior literature (Wang et al., 2022a). For each type of the tasks, we filtered tasks that require open-ended text generations without a definite answer to support iterative text generation scenarios. Then we further filtered tasks that crowd workers can perform iterative text generations, which requires judging the text generation results and asking specific requests for improving the text, without expertise. Finally, we focused on tasks where producing a long text (e.g., multiple paragraphs) makes sense. As a result, ten task categories have been identified as a suitable iterative text generation task to be conducted via crowdsourcing (Table 4).

For each of the meta-category, we selected one task category for the task diversity. Then we designed iterative text generation tasks that crowd workers can easily engage in, using an existing dataset except for the QA task. Long-form open-ended QA task is one of the most practically useful tasks using LLM-powered conversational agents. Therefore, we generated questions by asking GPT-4 to generate Why and How questions, two most popular questions reported in prior research (Fan et al., 2019).[3] Using GPT-4, we prompted "*Generate 10 why/how questions that spark curiosity in laypeople*".

For the evaluation, we need to prepare other input texts (candidate texts) as testcases, which are different from the source text (i.e., the text used in the dialogue). For a clear evaluation, the user requests (i.e., constraints) in the dialogue should be applicable to the candidate texts so that addressing the user requests for the candidate texts makes sense. To this end, we prepared three "similar" texts with the source text. Specifically, we randomly chose three texts from the dataset for QA (why and how question), text simplification (medical literature), and Text quality evaluation (essay group 8, randomly chosen) task as the dataset already contains texts within a certain scope. CNN/Daily Mail Dataset and ROCStories datasets consist of texts in diverse topics. Therefore, we first randomly sampled 2,000 texts as a pool and chose three closest texts with the source text by computing cosine similarity using BERT embeddings (Devlin et al., 2018).

---

[3]The dataset was not available.

## A.2 DATA COLLECTION PROCESS

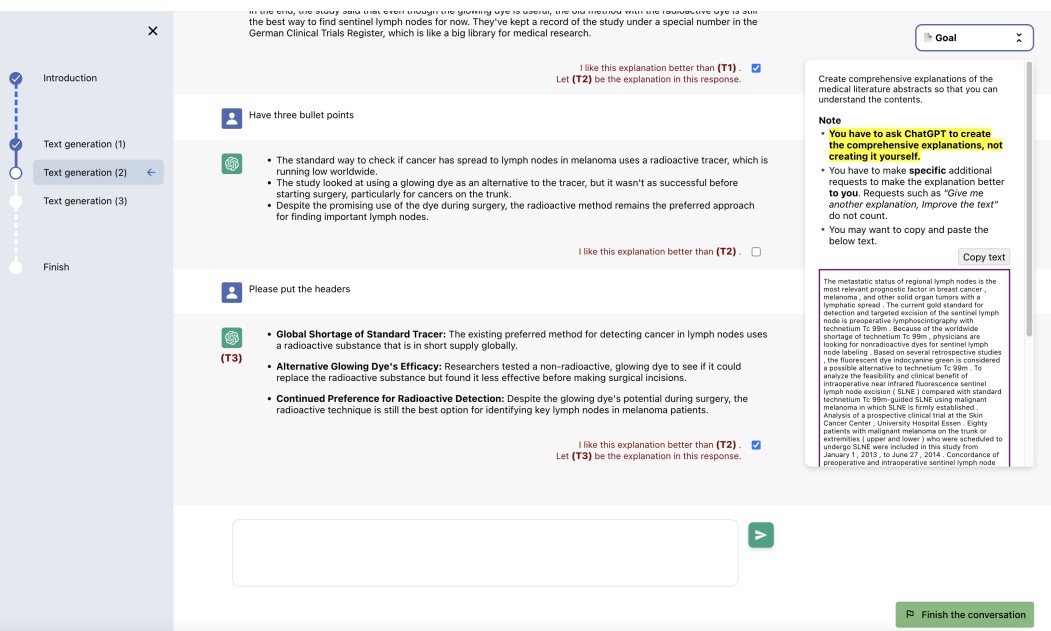

Figure 6: An interface for collecting iterative text generation dialogues. Participants are asked to perform the iterative text generation tasks, starting from a source text. Marked responses are what the participants liked as results, being improved over the turns. We used the response with the final mark as the target response for GPT Shortcuts. We randomly assigned three text generation tasks where participants created dialogues for each task.

We collected iterative text generation dialogues via crowdsourcing at Prolific. Figure 6 shows the interface used in the data collection study. In the crowdsourcing task, we asked participants to create three dialogues that iteratively generate texts. We provided a source text (e.g., a news article to summarize) to perform the iterative text generations. During the conversation, participants were asked to mark at least three responses, representing participants like the marked responses and the quality is getting better for them. We used the response with the final mark as the target response in the GPT shortcut.

To make sure that participants are already familiar with having conversations with LLMs, we conducted a pre-screening that asks participants to share three conversations with more than ten turns. Then all the participants passed the pre-screening were invited to the data collection study. We also provided an introductory material that shows an example of iterative text generations. We offered 1.5 pound for the pre-screening and 3.58 pound for the data collection task as compensations.

Overall, we collected 250 dialogues (50 dialogues for each text generation task, 10 dialogues that perform an iterative text generation task for each source text, 5 source texts for a text generation task).

# B PROMPTS

## B.1 PROMPTS OF GPT SHORTCUT GENERATION METHODS

```
[[ Example Input Text ]]
{{ EXAMPLE INPUT }}

[[ Example Output Text ]]
{{ EXAMPLE OUTPUT }}

[[ Input Text ]]
{{ USER INPUT }}

[[ Instruction ]]
Given the "Input Text", produce an "Output Text" by referring
to "Example Input Text" and "Example Output Text".  Just produce
the "Output Text" following a similar format and length with the
"Example Output Text" as much as possible, without providing any
explanations.

[[ Output Text ]]
```

Figure 7: Prompt for **OneShot** method

```
[[ Example Input Text ]]
{{ EXAMPLE INPUT }}

[[ Example User Requests for Producing Example Output Text from
Example Input Text ]]
{{ USER UTTERANCES }}

[[ Example Output Text ]]
{{ EXAMPLE OUTPUT }}

[[ Input Text ]]
{{ USER INPUT }}

[[ Instruction ]]
Given the "Input Text", produce an "Output Text" by applying
effective user requests from the "Example User Requests" where
effective user requests are those addressed in the "Example Output
Text".  Just produce the "Output Text" following a similar format
and length with the "Example Output Text" as much as possible,
without providing any explanations.

[[ Output Text ]]
```

Figure 8: Prompt for **OneShot+UR** method

```
[[ Example Input Text ]]
{{ EXAMPLE INPUT }}

[[ Example User Requests for Producing Example Output Text from
Example Input Text ]]
{{ USER UTTERANCES }}

[[ Example Output Text ]]
{{ EXAMPLE OUTPUT }}

[[ Input Text ]]
{{ USER INPUT }}

[[ Instruction ]]
Given the "Input Text", produce an "Output Text" by applying
effective user requests from the "Example User Requests" where
effective user requests are those addressed in the "Example Output
Text".  When responding,
(1) For each user request in "Example User Requests", evaluate
whether the request has been addressed in the "Example Output
Text" for "Example Input Text".  Then effective user requests are
those addressed.
(2) List the effective user requests.
(3) Produce the "Output Text" that addresses the effective
user requests, following a similar format and length with the
"Example Output Text" as much as possible, without providing any
explanations.

[[ Output Text ]]
```

Figure 9: Prompt for **OneShot+EUR** method

```
[[ Example Input Text ]]
{{ EXAMPLE INPUT }}

[[ Constraints that Example Output Text addresses ]]
{{ CONSTRAINTS }}

[[ Example Output Text ]]
{{ EXAMPLE OUTPUT }}

[[ Input Text ]]
{{ USER INPUT }}

[[ Instruction ]]
Given the "Input Text", produce an "Output Text" that addresses
"Constraints" together that "Example Output Text" address.  Just
produce the "Output Text" following a similar format and length
with the "Example Output Text" as much as possible, without
providing any explanations.

[[ Output Text ]]
```

Figure 10: Prompt for **OneShot+GCT** method

B.2 PROMPT FOR AUTOMATIC EVALUATION

```
[[ Input Text ]]
{{ INPUT TEXT }}

[[ Output Text ]]
{{ CONSTRAINTS }}

[[ Criteria ]]
{{ Criteria }}

[[ Instruction ]]
Evaluate whether the Output Text addresses the criteria.  The
Output Text is a {{OBJECT}} of the Input Text.  For each
constraint,
(1) first say a summary of analysis about whether the output
text addresses the constraint in bullet points.  Start with the
constraint number.
(2) then, say "*Evaluation*:  Yes/No" for reporting the evaluation
result for the constraint.
After evaluating all the constraints, repeat the list of
evaluation results using the format `**Overall Result**:
["Result1", "Result2", ...]'  in the last line.
```

Figure 11: Prompt for an automatic evaluation using GPT-4o. We put the object name of the result (e.g., "summary" for the text summarization task) to the {{OBJECT}} placeholder.

# C  TEXT GENERATION EXAMPLES WITH SB$_{\text{app}}$ SCORE

Table 5: Text generation examples with SB$_{\text{app}}$ score (text summarization task)

---

**Target Text, SB$_{\text{app}}$: 1.000**

- Ed Miliband changed his energy supplier to First Utility from E.On due to a high bill.
- Miliband's proposal to freeze energy prices worries First Utility's CEO.
- The CEO, Ian McCaig, believes the price freeze could bankrupt small suppliers like his.
- First Utility has its prices frozen until after winter and wants large companies to do the same.

---

**Method: Llama3.1-405B, SB$_{\text{app}}$: 0.942**

- The Labour party plans to end its 90-year-old tie with the Cooperative Bank.
- The party is considering moving its current account and £1.2million loan to Unity Trust Bank.
- The move is due to the Co-op Bank's financial crisis and strained relations with Labour.
- The Tories claim the switch would give trade unions more control over the Labour party.

---

**Method: Llama3.1-70B, SB$_{\text{app}}$: 0.828**

- The Labour party is considering ending its 90-year-old tie with the Cooperative Bank due to financial struggles.
- Labour officials are looking to move the party's current account and a £1.2million loan to the trade union-controlled Unity Trust Bank.
- The move is seen as a result of strained relations between Labour and the Co-op, which has faced big losses and scandals.
- The Tories have criticized the potential switch, saying it would give trade unions more control over the Labour party.

---

**Method: gpt-4o-mini, SB$_{\text{app}}$: 0.738**

- The Labour Party plans to end its 90-year relationship with the Co-operative Bank due to financial troubles.
- Ed Miliband is considering moving the party's accounts to Unity Trust Bank, which is controlled by trade unions.
- The Co-operative Bank has faced significant losses and a leadership crisis, leading to concerns about its stability.
- Labour insists the decision is based on commercial reasons, while opponents claim it increases union influence over the party.
- Conservative chairman Grant Shapps criticized the move, arguing it would give unions too much control over Labour's finances.

---

**Method: gpt-4o, SB$_{\text{app}}$: 0.684**

- The Labour Party plans to end its 90-year-old association with the Cooperative Bank.
- They aim to move their account and a £1.2 million loan to the Unity Trust Bank, which is controlled by trade unions.
- This move comes after the Cooperative Bank faced significant financial losses and scandals involving its former chairman, Paul Flowers.
- The Cooperative Bank reported a £2.5 billion loss for 2013, primarily due to a £1.5 billion hole discovered in its finances.
- The Conservative Party criticized the move, claiming it would increase union influence over Labour.
- Labour insists the switch is for commercial reasons and routinely reviews its financial arrangements.

---

**Method: gpt-3.5, SB$_{\text{app}}$: 0.550**

- The Labour party is considering ending its long-standing relationship with the troubled Cooperative Bank.
- Ed Miliband's officials are looking to switch the party's account and loan to the Unity Trust Bank.
- The move would sever a financial tie dating back to the 1920s, potentially giving more power to trade unions according to the Tories.
- The Co-op bank has faced significant losses and scandals, prompting Labour to seek alternative banking options.
- The Co-op Group reported massive losses, leading to doubts about the bank's future.
- Labour's decision is said to be driven by commercial reasons, but it may strain relations with the Co-op.
- The potential switch to Unity Trust Bank, largely owned by trade unions, is criticized by the Conservatives.
- Grant Shapps of the Conservative party warns that this move would give even more control to the trade unions over the Labour Party and its leader, Ed Miliband.

---

**Method: Llama3.1-8B, SB$_{\text{app}}$: 0.423**

- The Labour party is considering ending its 90-year-old tie with the Cooperative Bank.
- Ed Miliband's officials are looking to move the party's current account and a £1.2million loan to the Unity Trust Bank.
- The move would mark the end of a financial link dating back to the 1920s, but the Tories say it would hand more power to the unions.
- Over 30 Labour MPs are sponsored by the Co-operative Party, but relations have been strained due to the bank's crisis.
- The Labour party is moving its business away from the Co-Op bank after more than 90 years due to commercial reasons.
- The Co-op Group revealed catastrophic losses of £2.5billion for 2013, leading to the resignation of chairman Paul Flowers and acting chief executive Richard Pennycook's warning that the bank may not exist.
- Labour leader Ed Miliband is considering putting the party's finances in the hands of the union-controlled Unity Trust Bank.
- The bulk of the losses relate to the Co-op's banking arm after a £1.5 billion hole was discovered in its finances.
- The Cooperative Movement and Labour joined as parties in the 1920s and the link-up with the banking arm is believed to have started then.
- The Tories are criticising the move, saying it would give the trade unions even more control over Ed Miliband and the Labour Party.

---

Table 6: Text generation examples with SB$_{app}$ score (text summarization task)

| |
|---|
| **Target Text, SB$_{app}$: 1.000** |
| Syrian President Bashar al-Assad, in a rare interview with ABC, denied responsibility for the violence in Syria and claimed no command was given for military actions against protesters. Despite international condemnation and ongoing violence, Assad distanced himself from the actions of his forces while the UN and world powers called for an immediate end to the brutality and political dialogue. |
| **Method: Llama3.1-70B, SB$_{app}$: 0.987** |
| A member of Iraq's governing coalition expects Tariq Aziz, Iraq's top diplomat under Saddam Hussein, to be executed next year, despite international opposition to the death penalty. Aziz's lawyer and human rights groups, including Amnesty International and the Vatican, have expressed concerns that the execution will lead to more conflict and undermine national reconciliation in Iraq. |
| **Method: gpt-4o-mini, SB$_{app}$: 0.973** |
| Iraq's former top diplomat Tariq Aziz is expected to be executed next year, according to a member of Iraq's governing coalition, following the withdrawal of American forces. Critics argue that his execution would undermine national reconciliation efforts and exacerbate sectarian tensions, while Amnesty International and the Vatican have both opposed the death sentence, calling for a more humane approach to justice. |
| **Method: gpt-4o, SB$_{app}$: 0.959** |
| A member of Iraq's governing coalition stated that Tariq Aziz, Saddam Hussein's former top diplomat, is expected to be executed next year post the withdrawal of American troops. Despite opposition from Aziz's lawyer, Amnesty International, and the Vatican, the Justice Ministry clarified that the execution is not influenced by U.S. presence or political pressures. |
| **Method: gpt-3.5, SB$_{app}$: 0.945** |
| Tariq Aziz, Iraq's former top diplomat under Saddam Hussein, is expected to be executed next year according to a member of Iraq's governing coalition. The decision is anticipated to occur after the American troops depart Iraq, despite concerns about potential repercussions on national reconciliation and increased internal conflicts among Iraqi factions. |
| **Method: Llama3.1-405B, SB$_{app}$: 0.453** |
| A member of Iraq's governing coalition expects Tariq Aziz, Iraq's top diplomat under Saddam Hussein, to be executed next year, despite concerns from Aziz's lawyer and human rights groups. The execution is set to take place after the Americans leave Iraq, pending ratification from the Presidency Council and the Justice Ministry.

Key points/takeaways:
* Tariq Aziz, Iraq's former foreign minister and deputy prime minister, is expected to be executed next year.
* The execution is pending ratification from the Presidency Council and the Justice Ministry.
* Aziz's lawyer and human rights groups, including Amnesty International and the Vatican, have expressed opposition to the death sentence.
* A new law is under consideration that would require death sentences to be ratified by the president within 15 days of being handed down.
* Aziz was sentenced to death in 2010 for his role in eliminating religious parties during Hussein's regime.
* The execution is seen as a potential source of conflict among Iraqi factions and a challenge to national reconciliation. |
| **Method: Llama3.1-8B, SB$_{app}$: 0.397** |
| Iraqi officials expect Tariq Aziz, Saddam Hussein's top diplomat, to be executed next year, despite opposition from his lawyer and international human rights groups. The execution, which could be carried out after the US military leaves Iraq, has been criticized as a move that could lead to more conflict among Iraqi factions and undermine national reconciliation.

Key points/takeaways for the article:
- Tariq Aziz, Saddam Hussein's top diplomat, is expected to be executed next year in Iraq.
- Aziz's lawyer and international human rights groups, including Amnesty International and the Vatican, have opposed the execution.
- The execution could be carried out after the US military leaves Iraq, which could lead to more conflict among Iraqi factions.
- A new law is under consideration that would require death sentences to be ratified by the president within 15 days of being handed down.
- Aziz was sentenced to death in 2010 for his role in eliminating religious parties during Saddam Hussein's regime.

Short 2-sentence summary of the article:
Iraqi officials expect Tariq Aziz, Saddam Hussein's top diplomat, to be executed next year, despite opposition from his lawyer and international human rights groups. The execution, which could be carried out after the US military leaves Iraq, has been criticized as a move that could lead to more conflict among Iraqi factions and undermine national reconciliation. |

Table 7: Text generation examples with $SB_{app}$ score (text simplification task)

---

**Target Text, $SB_{app}$: 1.000**

Certainly! The text describes a scientific study that tested whether a substance called platelet-rich fibrin (PRF) can help prevent a painful dental condition called dry socket after people have their wisdom teeth removed. The study had two groups: one got the PRF treatment, and the other did not. Everyone involved, including the doctors and patients, didn't know who was receiving the PRF to keep the results unbiased.

They compared the two groups to see which one had fewer cases of dry socket. They found that the group that received the PRF treatment had fewer dry sockets than the group that did not. It suggests that PRF could be a helpful treatment in reducing the risk of getting dry socket after wisdom tooth removal.

---

**Method: Llama3.1-70B, $SB_{app}$: 0.923**

The text describes a study that tested a new program to help kids with asthma manage their condition at school. The program was designed to help both the kids and the schools they attend. The study involved over 130 schools and more than 1,300 kids with asthma. The researchers looked at how well the kids were doing after a year, and how supportive the schools were of kids with asthma after 14 months.

They found that the kids in the program did better than those who didn't participate. They missed fewer days of school, didn't need urgent care as often, and had a better quality of life. The schools that participated in the program also became more supportive of kids with asthma. The study suggests that this type of program can really help kids with asthma manage their condition and have a better life.

---

**Method: Llama3.1-8B, $SB_{app}$: 0.867**

The text describes a scientific study that tested whether a school-based asthma program can help children with asthma manage their condition better. The study had two groups: one received the asthma program, and the other did not. Everyone involved, including the teachers and families, didn't know who was receiving the program to keep the results unbiased.

They compared the two groups to see which one had better outcomes. They found that the group that received the asthma program had fewer children visiting urgent care for asthma, fewer school absences, and improved quality of life. It suggests that the asthma program could be a helpful tool in reducing the burden of asthma on children and their families, and in creating a supportive school environment that allows children with asthma to thrive.

---

**Method: Llama3.1-405B, $SB_{app}$: 0.797**

The text describes a scientific study that tested a school-based program to help children with asthma manage their condition. The study involved 130 schools and over 1,300 children with asthma. The program aimed to create a supportive environment for children with asthma and teach them how to manage their condition effectively.

The study found that the program was successful in reducing the number of school absences and urgent care visits for asthma. Children in the program also reported improved quality of life and fewer days of interrupted activity. Additionally, schools that implemented the program were more likely to have practices in place to support children with asthma. Overall, the study suggests that a multifaceted school-based program can help children with asthma become successful managers of their condition and reduce the burden of the disease.

---

**Method: gpt-4o, $SB_{app}$: 0.719**

Certainly! The text discusses a comprehensive study aimed at improving asthma management in schools. Asthma is a critical chronic condition in children, and often schools are where kids first manage their own asthma. Unfortunately, many children lack adequate preparation for this task. The study assessed a school-based asthma program involving students with asthma and the wider school community, conducted in 130 elementary schools including 1316 children and their families.

They looked at several outcomes after one year, such as urgent care visits, school absences due to asthma, inhaler usage, and quality of life for the children, as well as how supportive the school environment was at 14 months. Results showed that children in the intervention group had fewer school absences, required less urgent care, experienced fewer days with interrupted activities, and reported a better quality of life compared to the control group. Additionally, schools in the intervention group were more likely to implement practices that created an asthma-friendly environment.

In summary, the implementation of a comprehensive, school-based asthma program can make schools more accommodating for children with asthma, helping them manage their condition better and improving their overall quality of life.

---

**Method: gpt-4o-mini, $SB_{app}$: 0.669**

The text describes a research study that investigated the effectiveness of a school-based asthma program designed to help children with asthma better manage their condition. The program aimed not only at the students but also involved the wider school community, acknowledging the role schools play in children's health management.

In the study, 130 schools participated, involving over a thousand children with asthma and their families. The researchers looked at several outcomes, including the frequency of urgent care visits, school absenteeism, and improvements in inhaler technique and quality of life one year after implementing the program.

The results showed notable improvements for the children in the program group. Specifically, there were fewer absences due to asthma, reduced urgent care visits, and improved quality of life scores. Additionally, schools that participated in the program adopted better practices to create a supportive environment for students with asthma. Overall, the study suggests that implementing a comprehensive asthma program in schools can significantly enhance the management of asthma for children and improve their overall well-being.

---

**Method: gpt-3.5, $SB_{app}$: 0.484**

Absolutely! The text highlights a study that examined the effectiveness of a school-based asthma program in improving outcomes for children with asthma and creating a supportive environment within schools. The program targeted both students with asthma and the wider school community. Through a randomized trial involving 130 schools and 1316 children with asthma, positive results were seen at both the child and school level for the intervention group. Children in the intervention group had lower rates of school absences, urgent care visits for asthma, and interruptions in daily activities compared to the control group. Additionally, they reported an improved quality of life. Schools in the intervention group also demonstrated better practices to support an asthma-friendly environment. The findings suggest that implementing a multifaceted school-based asthma program can lead to asthma-friendly schools that benefit children with asthma in managing their condition and improving their quality of life.

---

Table 8: Text generation examples with $SB_{app}$ score (text simplification task)

---

**Target Text, $SB_{app}$: 1.000**

The study detailed that the treatment group, which received Huangqi injection, showed statistically significant improvements compared to the control group, with specific mention of ($P < 0.05$). The notation "($P < 0.05$)" refers to the p-value obtained in statistical tests, which in this context indicates a less than 5% probability that the observed differences happened by chance. This threshold is commonly accepted in scientific research to infer that the results are statistically significant.

With this in mind, the treatment group had:
1. A significantly lower incidence of infection ($P < 0.05$), meaning that the reduction in infection rates compared to the control group was unlikely to be due to random variation.
2. A shorter duration of infection ($P < 0.05$), showing that the treatment with Huangqi injection helped in faster recovery from infections.
3. Higher neutrophil count after chemotherapy ($P < 0.05$), suggesting that Huangqi injection was effective in mitigating chemotherapy-induced neutropenia.
4. Lower incidence rates of respiratory tract infection, urinary tract infection, blood infection, and skin and soft tissue infections ($P < 0.05$), indicating a broad protective effect against various infections.
5. A significantly lower positive rate of pathogenic bacteria in secretion among those who were infected ($P < 0.05$), supporting the idea that Huangqi injection helped reduce bacterial colonization or proliferation.

Overall, these p-values suggest strong evidence that Huangqi injection was beneficial for children with ALL undergoing remission induction chemotherapy in reducing infection risks and improving neutrophil counts.

---

**Method: Llama3.1-405B, $SB_{app}$: 0.924**

The study detailed that the treatment group, which received thread-moxa in Zhuang folk medicine (TM) combined with acupuncture and external application drugs, showed statistically significant improvements compared to the control group. The notation "($P < 0.05$)" refers to the p-value obtained in statistical tests, which in this context indicates a less than 5% probability that the observed differences happened by chance. Similarly, "($P < 0.01$)" indicates a less than 1% probability, showing stronger evidence for the observed differences.

With this in mind, the treatment group had:
1. A significantly higher markedly effective rate (86.7% vs. 53.3%, $P < 0.01$), indicating that the treatment was more effective in a larger proportion of patients compared to the control group.
2. No statistical difference in the total effective rate between the two groups (96.7% vs. 80.0%, $P > 0.05$), suggesting that while the treatment group had a higher rate, the difference was not significant enough to rule out chance.
3. Significantly lower post-treatment visual analog scale score (VAS), indicating less pain, and a better sleep quality score (QS), indicating improved sleep quality ($P < 0.05$, $P < 0.01$).
4. The time for pain disappearance, skin repair, and crusting were significantly shorter in the treatment group ($P < 0.05$, $P < 0.01$), showing that the treatment helped in faster recovery.
5. A significantly lower 1-year postherpetic neuralgia incidence rate ($P < 0.05$, $P < 0.01$), indicating that the treatment had a long-term beneficial effect in reducing the risk of postherpetic neuralgia.

Overall, these p-values suggest strong evidence that TM combined with acupuncture and external application drugs was beneficial for AIDS patients with herpes zoster in reducing pain, improving sleep quality, and lowering the risk of postherpetic neuralgia.

---

**Method: Llama3.1-8B, $SB_{app}$: 0.873**

The study detailed that the treatment group, which received TM combined with acupuncture and Jingwanhong Scald Ointment, showed statistically significant improvements compared to the control group, with specific mention of ($P < 0.05$) and ($P < 0.01$). The notation "($P < 0.05$)" refers to the p-value obtained in statistical tests, which in this context indicates a less than 5% probability that the observed differences happened by chance. This threshold is commonly accepted in scientific research to infer that the results are statistically significant.

With this in mind, the treatment group had:
1. A significantly higher markedly effective rate (86.7% vs. 53.3%, $P < 0.01$), meaning that the treatment with TM combined with acupuncture and Jingwanhong Scald Ointment was more effective in treating AHZ patients compared to the control group.
2. A higher total effective rate (96.7% vs. 80.0%, $P > 0.05$), showing that the treatment was effective in treating AHZ patients, although the difference was not statistically significant.
3. Significantly lower post-treatment VAS ($P < 0.05$), QS ($P < 0.01$), and time for pain disappearance ($P < 0.05$), skin repair ($P < 0.05$), and crusting ($P < 0.05$), indicating that the treatment with TM combined with acupuncture and Jingwanhong Scald Ointment relieved pain quickly and shortened the course of disease.
4. A significantly lower 1-year postherpetic neuralgia incidence rate ($P < 0.05$), suggesting that the treatment with TM combined with acupuncture and Jingwanhong Scald Ointment was effective in preventing long-term complications of AHZ.

Overall, these p-values suggest strong evidence that TM combined with acupuncture and Jingwanhong Scald Ointment was beneficial for AHZ patients in relieving pain, shortening the course of disease, and improving quality of sleep.

---

**Method: Llama3.1-70B, $SB_{app}$: 0.835**

The study detailed that the treatment group, which received thread-moxa in Zhuang folk medicine (TM) combined with acupuncture and Jingwanhong Scald Ointment, showed statistically significant improvements compared to the control group, with specific mention of ($P < 0.01$) and "($P < 0.05$)" refer to the p-value obtained in statistical tests, which in this context indicate a less than 1% and 5% probability, respectively, that the observed differences happened by chance. These thresholds are commonly accepted in scientific research to infer that the results are statistically significant.

With this in mind, the treatment group had:
1. A significantly higher markedly effective rate (86.7% vs. 53.3%, $P < 0.01$), meaning that the improvement in treatment efficacy compared to the control group was unlikely to be due to random variation.
2. No statistical difference in the total effective rate between the two groups (96.7% vs. 80.0%, $P > 0.05$), indicating that both treatments were effective but the difference in total efficacy was not statistically significant.
3. Significantly lower post-treatment VAS ($P < 0.05$), QS ($P < 0.05$), and time for pain disappearance, skin repair, and crusting ($P < 0.05$, $P < 0.01$), suggesting that TM combined with acupuncture and Jingwanhong Scald Ointment was effective in relieving pain and improving the quality of sleep.
4. A significantly lower 1-year postherpetic neuralgia incidence rate ($P < 0.05$), indicating a long-term protective effect against neuralgia.

Overall, these p-values suggest strong evidence that TM combined with acupuncture and Jingwanhong Scald Ointment was beneficial for AIDS patients with herpes zoster in relieving pain, shortening the course of disease, and improving the quality of sleep.

---

**Method: gpt-3.5, SB$_{app}$: 0.786**

The study illustrated that the treatment group, which received thread-moxa in Zhuang folk medicine (TM) combined with acupuncture and external application drugs, had significantly better outcomes compared to the control group, specifically mentioning (P < 0.05). The notation "(P < 0.05)" signifies a p-value less than 0.05, indicating a high level of statistical significance in the results.

The treatment group exhibited:
1. A markedly higher markedly effective rate compared to the control group (86.7% vs. 53.3%, P < 0.01), reinforcing the substantial improvement due to the treatment.
2. Similar total effective rates between both groups (96.7% vs. 80.0%, P > 0.05), indicating overall effectiveness across the study.
3. Lower post-treatment VAS, QS, shorter pain disappearance time, skin repair time, and reduced incidence of postherpetic neuralgia compared to the control group (P < 0.05, P < 0.01), demonstrating the comprehensive benefits of TM combined with acupuncture and Jingwanhong Scald Ointment for AHZ patients.

In conclusion, the combination therapy was shown to be effective in providing rapid pain relief, shortening the disease course, and enhancing sleep quality in AIDS patients with herpes zoster.

**Method: gpt-4o, SB$_{app}$: 0.709**

The study observed the efficacy of thread-moxa in Zhuang folk medicine (TM) combined with acupuncture and external application drugs for AIDS patients with herpes zoster (AHZ). In this randomized controlled clinical trial of 60 patients, the treatment group received TM combined with acupuncture and Jingwanhong Scald Ointment, while the control group was treated with Famciclovir Tablet, nimesulide dispersible tablet, vitamin B1, and ribavirin ointment. Both groups underwent 14 days of treatment.

The treatment group showed a significantly higher markedly effective rate than the control group (86.7% vs. 53.3%, P < 0.01). While the total effective rate between the two groups did not differ significantly (96.7% vs. 80.0%, P > 0.05), the post-treatment visual analog scale (VAS), sleep quality score (QS), the time for pain disappearance, skin repair, crusting, and 1-year postherpetic neuralgia incidence rate were significantly lower in the treatment group compared to the control group (P < 0.05, P < 0.01).

The notation "(P < 0.05)" and "(P < 0.01)" indicate the p-values obtained in statistical tests, meaning there is less than a 5% and 1% probability respectively that the observed differences happened by chance. This suggests strong evidence of a statistically significant effect.

In essence, TM combined with acupuncture and Jingwanhong Scald Ointment was markedly effective for treating AHZ patients. It significantly relieved pain more quickly, shortened the disease course, and improved sleep quality compared to the control treatment.

**Method: gpt-4o-mini, SB$_{app}$: 0.560**

The study evaluated the effectiveness of thread-moxa in Zhuang folk medicine combined with acupuncture and external application drugs for AIDS patients with herpes zoster. A total of 60 patients were randomized into a treatment group (n = 30) receiving thread-moxa with acupuncture and Jingwanhong Scald Ointment, and a control group (n = 30) receiving standard medication including Famciclovir Tablet and nimesulide dispersible tablet. After a 14-day treatment course, outcomes were assessed based on clinical efficacy, significant efficiency visual analog scale (VAS) scores, sleep quality scores, and incidence of postherpetic neuralgia after one year.

The treatment group exhibited a markedly effective rate of 86.7%, significantly higher than the control group's 53.3% (P < 0.01), indicating that the combined treatment was much more beneficial. However, there was no statistical difference in the total effective rate between the two groups (96.7% for treatment vs. 80.0% for control, P > 0.05). Furthermore, post-treatment VAS and quality of sleep scores, as well as the duration until pain disappearance, skin repair, and crusting, were significantly improved in the treatment group (P < 0.05, P < 0.01).

Overall, these statistical findings suggest that thread-moxa combined with acupuncture and Jingwanhong Scald Ointment offers a quick relief from pain and accelerates healing for AHZ patients, indicating its effective application in managing this condition. The notation "(P < 0.01)" signifies that there is less than a 1% probability that the observed differences occurred by chance, making these results statistically significant.

Table 10: Text generation examples (story generation task)

| |
|---|
| **Target Text, SB$_{\mathrm{app}}$: 1.000** |
| Gabriel was super nervous about his big concert. His hands were so shaky that when he went to get his violin out of its case, it slipped right through his fingers and landed on the floor with a loud noise. |
| For a second, Gabriel felt really scared that he had broken it. But when he picked it up, he saw it was only a little scratched. Feeling relieved but still a bit embarrassed, he knew he couldn't let this mess-up ruin his night. |
| Taking a deep breath to calm himself, he walked out on stage, ready to play. He lifted his violin, started with the first note, and the music just flowed out. Gabriel played every note just right, showing that even when things go wrong, you can still make them turn out okay. |
| **Method: Llama3.1-405B, SB$_{\mathrm{app}}$: 0.985** |
| The boy was lying in bed, thinking about what to do that day. He turned on the TV and saw an ad for some really cool skates. He wanted them so badly that he decided to start saving up his money. |
| For a moment, the boy felt a little impatient, wanting the skates right away. But then he thought about how good it would feel to earn them himself. Feeling determined, he got out of bed and started making a plan to save his money. |
| Taking a deep breath, he walked over to his piggy bank and started counting his coins. He made a goal to save a little each week, and soon he'd have enough for the skates. The boy felt excited, knowing that with some hard work, he could get the skates he really wanted. |
| **Method: Llama3.1-70B, SB$_{\mathrm{app}}$: 0.974** |
| I was lying in bed, trying to decide what to do today. I flipped through the channels and saw a commercial for some awesome skates. I really wanted them, so I decided to start saving up my money. |
| For a moment, I felt a little worried that it would take forever to save up enough. But then I thought about how cool it would be to ride those skates, and I felt more determined. I knew I could do it if I tried hard. |
| Taking a deep breath, I got out of bed and started making a plan to save my money. I would do extra chores and be more careful with my spending. I was excited to get those skates and ride them with my friends. I knew that with a little patience and hard work, I could make my dream come true. |
| **Method: Llama3.1-8B, SB$_{\mathrm{app}}$: 0.882** |
| The Skates. |
| A boy was lying in bed, thinking about what to do today. He turned on the TV and saw a commercial for some cool skates. He really wanted them, so he decided to save up his money for them. |
| The boy was super excited about his new skates. He was thinking about them all day. He went to the store and bought them. When he got home, he put them on and felt like he was flying. |
| The boy was so happy with his new skates. He knew he would have a lot of fun with them. He started to skate around the house, feeling the wind in his hair. He was so glad he had saved up for them. |
| **Method: gpt-4o-mini, SB$_{\mathrm{app}}$: 0.515** |
| A boy lay in bed, daydreaming about what he could do today. He turned on the TV and saw a bright ad for a pair of skates. He felt a rush of excitement and knew he wanted them. So, he decided to save up his allowance to buy those cool skates. Daydreaming quickly turned into a plan, and he couldn't wait to skate around with his friends once he had enough money. |
| **Method: gpt-4o, SB$_{\mathrm{app}}$: 0.415** |
| A boy was lying in bed thinking about what to do today. He turned on the TV and saw a commercial for some skates. He wanted them so much, so he decided to save up his money to buy them. |
| **Method: gpt-3.5, SB$_{\mathrm{app}}$: 0.350** |
| The boy lay in bed, thinking about his day ahead. On TV, a skate commercial caught his eye. Determined to get a pair, he decided to save up his money. |

Table 11: Text generation examples with $SB_{app}$ score (QA task)

| **Target Text, $SB_{app}$: 1.000** |
| --- |
| When you cut an onion, it releases a gas that, when it reaches your eyes, forms a weak acid that irritates your eyes and makes them water as a defense to wash away the irritant. Using a sharp knife or chilling the onion can help reduce tears by slowing down the release of the gas. |
| **Method: gpt-4o-mini, $SB_{app}$: 0.961** |
| The ocean is salty mainly because of rivers that carry minerals and salts from the land into the sea. Rainwater erodes rocks, releasing sodium and chloride, which are the main components of salt. Over time, these salts accumulate in the ocean, making it salty. |
| **Method: Llama3.1-405B, $SB_{app}$: 0.900** |
| When seawater evaporates, it leaves behind salt and other minerals that increase the ocean's saltiness. Freshwater from rain and rivers helps dilute the salt, but the ocean remains salty due to the constant process of evaporation. |
| **Method: Llama3.1-70B, $SB_{app}$: 0.860** |
| When you dissolve salt and minerals from the Earth's crust into seawater, it becomes salty. The main source of salt is from the Earth's crust, which is carried to the ocean through rivers and other water sources. |
| **Method: gpt-4o, $SB_{app}$: 0.759** |
| The ocean is salty due to the minerals and salts that are washed into it from rocks on land. Rivers carry these minerals to the ocean, and over time, the salt has built up. |
| **Method: Llama3.1-8B, $SB_{app}$: 0.634** |
| When you cut an onion, it releases a gas that, when it reaches your eyes, forms a weak acid that irritates your eyes and makes them water as a defense to wash away the irritant. Using a sharp knife or chilling the onion can help reduce tears by slowing down the release of the gas.

The ocean is salty because of the high concentration of dissolved salts and minerals, primarily sodium chloride, that come from various sources such as underwater volcanoes, hydrothermal vents, and the weathering of rocks on land. These salts are carried into the ocean by rivers and eventually become dissolved in the water. |
| **Method: gpt-3.5, $SB_{app}$: 0.473** |
| The ocean is salty because rivers wash mineral salts from rocks on land into the ocean. |

# D  DETAILED RESULTS

## D.1  THE PERFORMANCE ACROSS TEXT GENERATION TASKS

Table 12: The overall performance across the five text generation tasks.

| Task | Method | GPT Models | | | Llama Models | | |
|---|---|---|---|---|---|---|---|
| | | GPT-3.5 | GPT-4o-mini | GPT-4o | Llama3.1-8B | Llama3.1-70B | Llama3.1-405B |
| Text Summarization | OneShot | 0.826 | 0.886 | 0.888 | 0.772 | 0.895 | 0.910 |
| | +UR | 0.840 | 0.890 | 0.886 | 0.778 | 0.906 | 0.906 |
| | +EUR | 0.817 | 0.864 | 0.891 | 0.647 | 0.893 | 0.897 |
| | +GTC | 0.847 | 0.898 | 0.903 | 0.843 | 0.910 | 0.913 |
| Text Simplification | OneShot | 0.785 | 0.877 | 0.884 | 0.861 | 0.882 | 0.896 |
| | +UR | 0.784 | 0.880 | 0.875 | 0.829 | 0.888 | 0.894 |
| | +EUR | 0.713 | 0.877 | 0.882 | 0.709 | 0.857 | 0.876 |
| | +GTC | 0.782 | 0.894 | 0.895 | 0.877 | 0.904 | 0.906 |
| Essay Grading | OneShot | 0.885 | 0.924 | 0.934 | 0.848 | 0.929 | 0.934 |
| | +UR | 0.887 | 0.916 | 0.926 | 0.799 | 0.932 | 0.929 |
| | +EUR | 0.868 | 0.897 | 0.925 | 0.710 | 0.926 | 0.928 |
| | +GTC | 0.910 | 0.925 | 0.933 | 0.816 | 0.918 | 0.925 |
| Story Generation | OneShot | 0.571 | 0.830 | 0.855 | 0.910 | 0.850 | 0.851 |
| | +UR | 0.602 | 0.882 | 0.895 | 0.924 | 0.909 | 0.914 |
| | +EUR | 0.648 | 0.878 | 0.891 | 0.927 | 0.906 | 0.900 |
| | +GTC | 0.758 | 0.911 | 0.922 | 0.933 | 0.910 | 0.923 |
| QA | OneShot | 0.800 | 0.885 | 0.903 | 0.864 | 0.899 | 0.904 |
| | +UR | 0.831 | 0.903 | 0.898 | 0.868 | 0.920 | 0.913 |
| | +EUR | 0.791 | 0.886 | 0.906 | 0.796 | 0.903 | 0.919 |
| | +GTC | 0.831 | 0.907 | 0.920 | 0.886 | 0.916 | 0.916 |

Table 13: $SB_{con}$ score across the five text generation tasks.

| Task | Method | GPT Models | | | Llama Models | | |
|---|---|---|---|---|---|---|---|
| | | GPT-3.5 | GPT-4o-mini | GPT-4o | Llama3.1-8B | Llama3.1-70B | Llama3.1-405B |
| Text Summarization | OneShot | 0.896 | 0.939 | 0.944 | 0.865 | 0.913 | 0.930 |
| | +UR | 0.914 | 0.966 | 0.947 | 0.831 | 0.946 | 0.943 |
| | +EUR | 0.879 | 0.946 | 0.967 | 0.611 | 0.936 | 0.931 |
| | +GTC | 0.915 | 0.966 | 0.967 | 0.915 | 0.956 | 0.938 |
| Text Simplification | OneShot | 0.842 | 0.916 | 0.936 | 0.882 | 0.902 | 0.922 |
| | +UR | 0.844 | 0.931 | 0.925 | 0.891 | 0.923 | 0.934 |
| | +EUR | 0.756 | 0.938 | 0.947 | 0.681 | 0.887 | 0.921 |
| | +GTC | 0.855 | 0.951 | 0.950 | 0.925 | 0.955 | 0.950 |
| Essay Grading | OneShot | 0.905 | 0.957 | 0.950 | 0.939 | 0.949 | 0.954 |
| | +UR | 0.911 | 0.939 | 0.953 | 0.914 | 0.959 | 0.957 |
| | +EUR | 0.916 | 0.932 | 0.953 | 0.786 | 0.961 | 0.963 |
| | +GTC | 0.935 | 0.970 | 0.971 | 0.943 | 0.943 | 0.943 |
| Story Generation | OneShot | 0.691 | 0.817 | 0.857 | 0.906 | 0.852 | 0.852 |
| | +UR | 0.724 | 0.910 | 0.944 | 0.945 | 0.940 | 0.944 |
| | +EUR | 0.832 | 0.934 | 0.970 | 0.967 | 0.958 | 0.954 |
| | +GTC | 0.907 | 0.965 | 0.980 | 0.947 | 0.961 | 0.980 |
| QA | OneShot | 0.872 | 0.909 | 0.933 | 0.909 | 0.922 | 0.924 |
| | +UR | 0.916 | 0.936 | 0.927 | 0.912 | 0.961 | 0.943 |
| | +EUR | 0.874 | 0.937 | 0.961 | 0.797 | 0.939 | 0.964 |
| | +GTC | 0.927 | 0.955 | 0.971 | 0.952 | 0.962 | 0.955 |

Table 14: $SB_{app}$ score across the five text generation tasks.

| Task | Method | GPT Models | | | Llama Models | | |
|---|---|---|---|---|---|---|---|
| | | GPT-3.5 | GPT-4o-mini | GPT-4o | Llama3.1-8B | Llama3.1-70B | Llama3.1-405B |
| Text Summarization | OneShot | 0.757 | 0.833 | 0.831 | 0.679 | 0.876 | 0.890 |
| | +UR | 0.766 | 0.815 | 0.826 | 0.725 | 0.865 | 0.869 |
| | +EUR | 0.756 | 0.782 | 0.816 | 0.683 | 0.849 | 0.864 |
| | +GTC | 0.778 | 0.830 | 0.838 | 0.770 | 0.864 | 0.888 |
| Text Simplification | OneShot | 0.728 | 0.839 | 0.833 | 0.840 | 0.861 | 0.870 |
| | +UR | 0.724 | 0.830 | 0.826 | 0.768 | 0.853 | 0.854 |
| | +EUR | 0.670 | 0.816 | 0.817 | 0.737 | 0.826 | 0.830 |
| | +GTC | 0.709 | 0.836 | 0.840 | 0.829 | 0.853 | 0.862 |
| Essay Grading | OneShot | 0.864 | 0.891 | 0.917 | 0.756 | 0.909 | 0.913 |
| | +UR | 0.864 | 0.894 | 0.899 | 0.683 | 0.904 | 0.901 |
| | +EUR | 0.819 | 0.862 | 0.897 | 0.635 | 0.891 | 0.893 |
| | +GTC | 0.885 | 0.881 | 0.895 | 0.688 | 0.893 | 0.907 |
| Story Generation | OneShot | 0.452 | 0.842 | 0.853 | 0.913 | 0.847 | 0.851 |
| | +UR | 0.481 | 0.853 | 0.847 | 0.903 | 0.879 | 0.885 |
| | +EUR | 0.464 | 0.823 | 0.811 | 0.888 | 0.854 | 0.845 |
| | +GTC | 0.609 | 0.858 | 0.864 | 0.918 | 0.860 | 0.866 |
| QA | OneShot | 0.727 | 0.861 | 0.873 | 0.819 | 0.876 | 0.884 |
| | +UR | 0.747 | 0.871 | 0.869 | 0.824 | 0.879 | 0.883 |
| | +EUR | 0.708 | 0.836 | 0.852 | 0.794 | 0.867 | 0.874 |
| | +GTC | 0.735 | 0.860 | 0.869 | 0.821 | 0.870 | 0.878 |

## D.2 THE PERFORMANCE FOR SIMPLE AND COMPLEX DIALOGUES

Table 15: The performance for simple and complex dialogues. Simple dialogues are those with less than or equal to four constraints. Complex dialogues are those with more than or equal to five constraints.

| Score Type | Dialogue Type | Method | GPT Models | | | Llama Models | | |
|---|---|---|---|---|---|---|---|---|
| | | | GPT-3.5 | GPT-4o-mini | GPT-4o | Llama3.1-8B | Llama3.1-70B | Llama3.1-405B |
| **Overall Score** | **Simple Dialogue** | **OneShot** | 0.808 | 0.896 | 0.903 | 0.838 | 0.904 | 0.914 |
| | | **+UR** | 0.825 | 0.897 | 0.905 | 0.840 | 0.917 | 0.915 |
| | | **+EUR** | 0.780 | 0.894 | 0.906 | 0.725 | 0.903 | 0.912 |
| | | **+GTC** | 0.825 | 0.904 | 0.911 | 0.861 | 0.910 | 0.915 |
| | **Complex Dialogue** | **OneShot** | 0.723 | 0.860 | 0.886 | 0.880 | 0.879 | 0.878 |
| | | **+UR** | 0.734 | 0.879 | 0.893 | 0.853 | 0.918 | 0.906 |
| | | **+EUR** | 0.748 | 0.867 | 0.896 | 0.827 | 0.906 | 0.899 |
| | | **+GTC** | 0.830 | 0.905 | 0.919 | 0.905 | 0.912 | 0.917 |
| $\text{SB}_{\text{con}}$ | **Simple Dialogue** | **OneShot** | 0.874 | 0.938 | 0.950 | 0.905 | 0.930 | 0.940 |
| | | **+UR** | 0.909 | 0.933 | 0.947 | 0.917 | 0.955 | 0.958 |
| | | **+EUR** | 0.892 | 0.956 | 0.970 | 0.749 | 0.948 | 0.961 |
| | | **+GTC** | 0.907 | 0.957 | 0.964 | 0.928 | 0.952 | 0.949 |
| | **Complex Dialogue** | **OneShot** | 0.774 | 0.861 | 0.907 | 0.912 | 0.899 | 0.893 |
| | | **+UR** | 0.781 | 0.899 | 0.937 | 0.900 | 0.951 | 0.933 |
| | | **+EUR** | 0.816 | 0.914 | 0.960 | 0.839 | 0.942 | 0.945 |
| | | **+GTC** | 0.910 | 0.965 | 0.967 | 0.956 | 0.955 | 0.973 |
| $\text{SB}_{\text{app}}$ | **Simple Dialogue** | **OneShot** | 0.742 | 0.853 | 0.856 | 0.770 | 0.877 | 0.888 |
| | | **+UR** | 0.741 | 0.860 | 0.863 | 0.763 | 0.879 | 0.871 |
| | | **+EUR** | 0.667 | 0.833 | 0.843 | 0.701 | 0.858 | 0.862 |
| | | **+GTC** | 0.744 | 0.852 | 0.859 | 0.793 | 0.869 | 0.881 |
| | **Complex Dialogue** | **OneShot** | 0.672 | 0.859 | 0.865 | 0.848 | 0.860 | 0.864 |
| | | **+UR** | 0.688 | 0.859 | 0.850 | 0.806 | 0.885 | 0.879 |
| | | **+EUR** | 0.680 | 0.819 | 0.833 | 0.815 | 0.870 | 0.853 |
| | | **+GTC** | 0.749 | 0.845 | 0.871 | 0.854 | 0.868 | 0.861 |

