# OpenReview forum: "GPT Shortcuts: Learning Iterative Text Generation Patterns from a Dialogue"
_ICLR.cc/2025/Conference — ICLR 2025 Conference Withdrawn Submission_

### Official Review · Reviewer_UA7n · 2024-10-16

**Soundness:** 3
**Presentation:** 3
**Contribution:** 3
**Rating:** 5
**Confidence:** 4

**Summary:**

The paper proposes the concept of GPT Shortcuts, which learn iterative generation patterns through dialogues and directly generate tailored text. Furthermore, the authors introduce a new benchmark, ShortcutBench, created through crowdsourcing and consisting of dialogues across five NLP tasks. They validate GPT Shortcuts on six LLMs using four different prompting techniques, finding that larger LLMs perform better than smaller ones and that specifying constraints is crucial for improving performance.

**Strengths:**

- Paper is well written and quite easy to follow
- Authors proposes GPT Shortcut which can generate tailored output without having multi-turn dialogues
- A new benchmark ShortcutBench is created, which might be useful for future investigation in this area

**Weaknesses:**

- Author do not provide information about crowdsourcing, e.g. #participants, their background. Also lack of statistic, e.g. percentage of coverage/generalizability
- I don't fully get your motivation, because I think in most cases, text generation collaborated with human is necessary. In such a way, human deliver their thoughts "step by step". The final output is heavily dependent on users' preference. In this case, how could GPT shortcut help?

**Questions:**

- l.139: In Figure 5, gpt-4o, gpt-4o-mini, Llama3.1-70B and Llama3.1-405B have almost same tendency, while gpt3.5 and Llama3.1-8B have similar tendency. But gpt3.5 should be even larger than Llama3.1-70B. How could you conclude that smaller LLMs underperforms larger LLMs (l.26)
- l.256: You mentioned that you only control the coverage and generalizability of generated checklists. How could make sure that the checklist are correct and not hallucinated?
- l.263: you use the same LLM (gpt-4o) for generation and judgment. I wonder if LLM itself can find errors from its generated text. (https://arxiv.org/abs/2310.01798 ; https://aclanthology.org/2024.findings-acl.826/)
- l.306: Is $SB_{app}$ really sufficient to measure the structure similarity? Or $SB_{app}$  even not necessary, as soon as LLMs can provide outputs with similar content

---

### Official Review · Reviewer_cTde · 2024-10-19

**Soundness:** 2
**Presentation:** 3
**Contribution:** 2
**Rating:** 5
**Confidence:** 4

**Summary:**

The paper proposes a new task to extract a pattern, a so-called shortcut, from example dialogues that specify the constraints to generate a target text. The LLM needs to derive the shortcut as an abstract pattern to be applied to other constraints in test dialogues to generate a target text. The authors created a benchmark dataset for the task and tested various LLMs and different prompts. They evaluate the LLMs with the prompts in two ways: 1) the coverage of the constraints that need to be represented in the target text, 2) the structural similarity between the generated text and the reference text. The authors discuss the performance in relation to the different models and the different prompts.

**Strengths:**

- presents a new challenge for shortcuts that potentially make prompt writing easier for non-experts
- created a benchmark data set for the task that contains more complex dialogues than comparable datasets
- provided results and analysis of a number of LLMs and the different prompts
- give insights in what prompts work and what LLMs

**Weaknesses:**

- observations from the results are rather generic, e.g. big models better than small models or that llama3.1-1B performs well for story writing and GPT generates too short stories
- the paper can be improved by a deeper discussion on what goes well and in which cases the models+prompts have difficulty. Central is the capability to create a shortcut that is sufficiently abstract and still semantically effective. So what are the generalizations that need to be made and when do the LLMs fail and when can they do this correctly. In the text you give one example of such a generalization: we revised a constraint “make DVDs antique”into“make a commonly used item in the past become an antique” —> more insight in how many of these cases in the data set, how abstract and how specific is the constraint in the target? Having insight in this in relation to what goes well and which cases all LLMs struggle would make the paper a lot stronger
- you use edit distance as a method for the structural match. Why not standard text generation measures such as BLUE, ROUGE, METEOR, BertScore?
- not clear what sequence of line length is and why this is a good measure for structural match
- GPT-o is used to evaluate the coverage of constraints but how well does this work? You only mention you take constraints mentioned 7 out of 10 times, but what constraints are these and which ones are rejected?

**Questions:**

- Line 325: “Finally, SBconv” —> Finally, SBapp
- explain in more detail what sequence of line length is and why this is a good way to evaluate structural similarity
- you already give detailed results in the introduction, line 131-147, while a lot of details still need to be explained. This should come later with the result sections.
- Many limitations are actually future work suggestions.

---

### Official Review · Reviewer_Bzy5 · 2024-11-03

**Soundness:** 2
**Presentation:** 2
**Contribution:** 1
**Rating:** 3
**Confidence:** 4

**Summary:**

The paper describes work on learning iterative patterns of user constraints made with interactions from chat interfaces to produce outputs that align with said patterns. The author/s motivate the task by stating that users may impose constraints that are not immediately obvious and can be based from timesteps of interacting with the chat interfaces. The author/s built a small benchmark named ShortcutBench for this task by crowdsourcing chat interactions on selected tasks including summarization and story generation. The author/s evaluated six LLMs (mostly from the Llama and GPT family) for the benchmark and used two evaluation to measure matches of target text to generated text: SBcon measures constraints addressed via a checklist and GPT4o judge and SBapp measures closeness of format from target text by user and generated text by GPT shortcut. Results showed that providing more explicit constraints allow the models to provided expected target results. However, my major concern here is that user constraints are dependent and made when on-the-fly and not outright. Thus, the way the LLMs are being evaluated with the constraints being given outright may not capture the true complexity of the task. Overall, the paper also needs major clarification, additional experiments and increasing the size of the benchmark, and stronger evaluation for the results to be reliable.

**Strengths:**

The task that the paper proposes, learning the patterns of iterative changes from a series of chat interface interactions, is interesting and indeed has practical approaches and realistic use. I agree that the emergence of user constraints on-the-fly to arrive at is a one concept that should be modeled from interactions. The paper overall is fairly easy to read to understand what the author/s are trying to achieve.

**Weaknesses:**

I have several concerns with the technicality and evaluation of the paper which I list down as follows:

1. Mismatch of expectations in motivation vs actual experiments → The paper is motivated by the fact that interactions of users to chat interfaces (ChatGPT, Llama, etc) span across domains such as writing a clinical report following specific diagnostic rules in health as an example. However, these types of interaction require experts as users of the chat interfaces and not just regular humans. Thus, I find some mismatch between examples to motivate the problem vs. what was actually collected and tested in ShortcutBench. In ShortCutBench, the interactions are quite short (averages 3-5 turns) which might not be the case for domain-specific interactions. It seems to me that the test cases for ShortCutBench are more controlled and boxed to specific tasks (e.g., summarization) than what we would observe from domains such as in health (nurses generating clinical reports), education (teachers generating content for classroom reading delivery), etc. Thus, reframing of expectations in both introduction and motivation is necessary to reduce mismatch.

2. Small benchmark → I think 250 test instances for the benchmark is quite small and need more instances for performance of the benchmark to converge. The author/s can also add splits like what was extracted from WildChats and put it in the benchmark itself as well as other works that mined chat interactions. Likewise, the paper needs to be thorough with the information covered by the benchmark particularly distribution of topic, tasks, user information, and granularity of its test instances.

3. Presentation of task and setup → Text generation methods in Section 5.1 need to be visualized for better understanding of differences and characteristics of the constraints presented to prompts. Moreover, some paragraphs are not at all clearly discussed. For example, "We put the ground-truth constraints in the checklists in the prompt, expecting to yield the ceiling performance of the other methods as it clearly informs correct constraints without any confusion” in what form? In what manner? Are these just appended in the prompt? How long is this additional information? In the appendix, only the template is shown.

4. Need for better framing of results → The results in Table 2 and 3 look very closed to each other by just a hair of points. It might be better to conduct statistical tests to ensure which one or two methods of prompting are actually informative given that you already have the ground truth constraints prompting as the ceiling. Moreover, I would not consider the ground truth constraints to attribute to the performances of the LLMs since the constraints are formed during iterative dialogue with the LLM and not something that is available outright.

5. Need more baseline and specialized models to evaluate → Aside from the benchmark being small, the paper could be strengthen by evaluating diverse LLMs both with optimized with general chat capabilities and task-category specialized models like CoEDIT (https://arxiv.org/abs/2305.09857) for rewriting texts which may capture to user constraints.

6. Lack of human evaluation → GPT4o judging related GPT model performance may induce some bias as with any LLM-to-LLM type of judging/evaluation. From this, I would strongly suggest having human evaluation of the same prompts that the GPT4o judge evaluated using a Likert-style metric in parallel with SBapp. This way, there is a stronger picture of the selected LLMs performance for the task.

**Questions:**

Templates aside, what form are the different types of user constraints added to the different prompting styles?
Also, please address questions from weakness section.

---

### Official Review · Reviewer_n5Eo · 2024-11-04

**Soundness:** 2
**Presentation:** 2
**Contribution:** 2
**Rating:** 3
**Confidence:** 3

**Summary:**

The paper formalizes the task of learning to generate text that follows certain patterns given examples of those patterns (and optionally natural language descriptions of the patterns). Identifying these patterns can allow a model to adhere to a user's constraints on attributes of the output without the user having to fully specify these attributes for every query. To evaluate the performance of models on this task, the paper presents a new dataset and evaluation framework, and benchmarks a number of large language models on the task.

The task is formalized as learning to produce an output text $y$ given some input text $x$. This amounts to inferring some transformation function $f(x, y)$, and the evidence for inferring this transformation in this context is the a given example $(x', y')$. This function is elicited from people through dialogue $d$. So, the task amounts to generating $f_d(x)$ given some dialogue $d$ that describes the transformation of some example input $x'$ to its corresponding output $y'$.

The dataset is collected by sampling 5 different input texts $x'$ for each of 5 text generation tasks, having 10 crowdworkers interact with a system to edit text according to their preferences for each $x'$, resulting in $5 \times 5 \times 10 = 250$ iterative text generation dialogues, each of which specifies a unique $f_d$. For evaluation, additional inputs $x_{test}$ are chosen, and a set of constraints is automatically inferred from the dialogue $d$. An LLM-as-a-judge approach is used to evaluate whether outputs on $x_{test}$ adhere to the automatically extracted constraints. An additional edit-distance-based appearance criterion is also used in evaluation.

The paper presents results from a number of different models in the GPT and Llama families, generally finding high constraint satisfaction performance (averages close to or above 90% of the identified constraints being met).

**Strengths:**

- The data collection protocol does not impose constraints, and instead appears to allow crowdworkers to specify constraints that they would like to see in the outputs, which does allow complex and nuanced preferences to be surfaced in interactions between users and a text generation model.
- The choice of tasks for data collection is carefully considered and covers a variety of text generation types.

**Weaknesses:**

- The organization and presentation of the paper is in general quite confusing, making many things hard to understand.
  - The introduction goes into the 4th page, with the list of contributions appearing only at the top of the 4th page, which is quite unusual (the number of pages isn't an issue, details follow)
  - There is a substantial discussion of the findings in the introduction, at a place where a lot of the concepts and terminology is not adequately introduced.
  - The idea of "constraints" is an important one through the paper, used for evaluation and to determine an "oracle" setting for model evaluation. However, there are no examples of these in the paper, making it difficult to understand the evaluation criteria. Are the annotations in Figure 1a ("Give me a tomato pasta recipe", "I need one portion.   Also, I have only one pan", etc.) constraints/input the model inferring constraints? It appears that the constraints are extracted by GPT-4o but what these look like might vary and it isn't clear what these constraints might look like.
- It appears that models already do very well at this task (most achieve over 90% constraint satisfaction). This raises the question of the intended role of the dataset. However, there are also other aspects of the benchmark construction that might be contributing to this.
  - 404: "As a possible reason, we observed that the example output is often self-explanatory. For example, LLM responses tend to contain an introductory statement that explains the text to generate, such as “Sure, here are the key points you should remember for your test!”, which clearly describes the necessary constraints."
    - This suggests that the responses produced by the LLM are used in entirety, including any repetition of the constraints specified in the interaction. In essence, this does the job of inferring the constraints (the primary challenge in this task) for the model (as the authors note in this quote).
    - Additionally, the constraints used for evaluation are those which are identified to be met by the target text (255: "It is important to note that not all the user-specified constraints in the dialogue have been addressed in the target text."; 298: "necessary constraints refer to a set of user-specified constraints that **have been addressed** in the target texts") that a user generates in interaction with an LLM (as an aside, it's not very clear what LLM is used to obtain the ground truth answers here either). So, we have a situation where an LLM is given some instructions by a user and follows some (and possibly not all) of these instructions. For evaluation, only the instructions that the LLM does follow are retained. This skews the task in favor of the LLM, since only constraints that have already been proven to be satisfied by an LLM are retained for evaluation. This might be a reason for the high performance across the board.
  - If this understanding is not accurate, it highlights the need for clearer presentation of the method.
 - 408: "However, OneShot significantly underperformed compared to other methods when the target text did not clearly imply the necessary constraints. For instance, the gap of SBcon scores between OneShot and +GTC was 4.5 times larger on average when the dialogue contained more than four constraints compared to fewer (Table 3)."
    - The text says "for instance", but it isn't clear how instances of the second type (which have >4 constraints) are related to those of the first type (that did not clearly imply the necessary constraints).
    - Scrubbing out explicit mentions of the constraints, or evaluating the drop in performance for instances where the constraints are not specified might reveal the task to be far more challenging that the numbers reported in the paper show.
- The appearance-based metric doesn't seem to be well-motivated. It mostly measures the degree to which the produced response adheres to the bullet point-like structure shown in the example. However, why this should be weighed equally with constraint satisfaction (which considers the more important question of what should be in the response) is not clear or well-argued.
  - Additionally, it isn't clear what is used as the target when computing this metric for held-out samples, since it's unclear whether ground-truth responses are collected for those.

**Questions:**

- Why the name "GPT shortcuts"? The idea of reusable sets of constraints/preferences for text generation could go beyond GPT models.

---

### Official Review · Reviewer_EJna · 2024-11-08

**Soundness:** 2
**Presentation:** 2
**Contribution:** 2
**Rating:** 5
**Confidence:** 4

**Summary:**

The paper studies the problem of adhering LLMs to domain specific requirements without respecifying the constraints. In this context some LLMs prompts are studied and analyzed. A dataset/ benchmark of such traces is provided which could be useful to the community.

**Strengths:**

- The problem is interesting and has practical value
- The dataset traces are very interesting
- The dataset itself is small but such traces are valuable (can I look at the dataset)

**Weaknesses:**

**Missing related work**: Given the vast literature/ work in LLM space, it is easy to miss related work, which could be now considered. Some examples:
- Memory augmented LMs [1, 4] where a memory can store user preferences and past feedback to align with this in the future. These class of methods work at inference time and do not update any model parameters, similar to this paper.
- Custom generation LMs e.g., [2] tailors how-to procedures to a user based on a dialogue. Given a goal and an uncustomized procedure P and a user's customization constraint H, they generate P′, a customized procedure that accounts for H. These H are quite similar in spirit to the constraints defined in the current paper under review. A benchmark over this dataset is available and might be valuable to compare in the current work. The techniques (GPT based, zero-shot and pipelines) share similarity to the ones described in Section 5 of the current paper. The evaluation metric (edit based MSED) also finds support in this paper.
- User assistant LMs e.g., OpenAI APIs [3] have a "GPT assistant" to create an agent instance to follow particular styles (similar to the motivation in this paper).


**Limited technical depth** e.g., in the "methodology" section there are essentially four prompts while ideas much of the related work progress can be used. I feel that the solution offered could be made much richer with a user memory that stores preferences for different tasks and how to structure/ represent and utliize this memory of traces is a research question. Secondly how the preference transfer happens between related tasks is the second research question. For example, given a user's preference in task A which is implicit in the revision trace, can we align to that preference for taskA in future inferences; or for task A'  (~A) if that preference can be transferred.

**Results**: The findings are good, but are not surprising: larger models can adhere to constraints; and when constraints are specified precisely it improves performance ([4] which had similar findings on understanding user feedback for iterative text generation). I also wanted to understand the kind of errors that the model makes in a bit more detail, but error analysis is missing.  The proposed metrics are interesting, but SB_{app} that resembles structural similarity of output and target text is a bit complex as a metric and not explained well. I support the intuition of this metric as other papers [2] have highlighted that refinement of a structure could drift too much from the original structure. I also wonder why an LLM-as-an-evaluator is not tried (or atleast as a second evaluator).

**Other minor points**: The prompts in the appendix B are a bit abstract. I was looking for one complete example. I tried but could not find the code and the dataset but could not find it.



References:
1. Memory-assisted prompt editing to improve GPT-3 after deployment ; EMNLP 2022; https://aclanthology.org/2022.emnlp-main.183/ ; (_augments LLM with a memory to store user preferences and past feedback or user preference_)
2. Tailoring with Targeted Precision: Edit-Based Agents for Open-Domain Procedure Customization ; ACL 2024;  https://aclanthology.org/2024.findings-acl.921.pdf (_custom how-to procedure generation with LLMs_)
3. GPT "assistants" ; https://help.openai.com/en/articles/8673914-gpts-vs-assistants (_in-context learning to personalize an agent_)
4. Self-Refine: Iterative Refinement with Self-Feedback ; NeurIPS 2023 (_LLM self-refinement through fine-grained feedback generated_)

**Questions:**

Please go through the weaknesses and clarify if there is something that I am missing.

---

### Author Response · Authors · 2024-11-19
**Thank you for reviewing our paper**

We appreciate the reviewers for the constructive feedback. We will reflect the suggestions and submit the paper to a future venue.

---

### Note · Authors · 2024-11-19

I have read and agree with the venue's withdrawal policy on behalf of myself and my co-authors.